# Stress experience and hormone feedback tune distinct components of hypothalamic CRH neuron activity

Joon S. Kim [1], Su Young Han[1] & Karl J. Iremonger [1]*

Stress leaves a lasting impression on an organism and reshapes future responses. However, the influence of past experience and stress hormones on the activity of neural stress circuits remains unclear. Hypothalamic corticotropin-releasing hormone (CRH) neurons orchestrate behavioral and endocrine responses to stress and are themselves highly sensitive to corticosteroid (CORT) stress hormones. Here, using in vivo optical recordings, we find that CRH neurons are rapidly activated in response to stress. CRH neuron activity robustly habituates to repeated presentations of the same, but not novel stressors. CORT feedback has little effect on CRH neuron responses to acute stress, or on habituation to repeated stressors. Rather, CORT preferentially inhibits tonic CRH neuron activity in the absence of stress stimuli. These findings reveal how stress experience and stress hormones modulate distinct components of CRH neuronal activity to mediate stress-induced adaptations.

¹ School of Biomedical Sciences, Centre for Neuroendocrinology and Department of Physiology, University of Otago, Dunedin 9054, New Zealand.
*email: karl.iremonger@otago.ac.nz

Stress is a collective term encompassing the repertoire of neural, endocrine, and physiological responses an organism mounts in the face of threat[1]. Stressful experiences evoke a cascade of hormonal and neural changes to promote adaptation and reshape future responses[2,3]. It has long been appreciated that neural circuits in the hypothalamus are essential for coordinating organism's responses to real or perceived threats[4–6]. One hypothalamic neural population, which are essential for controlling stress responses, are the corticotropin-releasing hormone (CRH) neurons located in the paraventricular nucleus (PVN). In addition to their well-known role in controlling corticosteroid (CORT) secretion, this neural population has also been shown to be important in mediating other stress-related functions, including shifts in behavior[7], pheromone release[8], and encoding of valence[9]. Furthermore, prior stress experience can modify subsequent behavioral and endocrine output[7,10–12], suggesting that CRH neuron responses may be highly adaptable.

Two broad forms of plasticity are thought to regulate CRH neuron responses to stress:[2,13,14] plasticity induced by activation of stress hormone receptors and plasticity induced by stress-evoked neural activity. Stress-induced CORT elevations have long been theorized to feedback to the brain and acutely "shut-off" CRH neuron activity and limit future stress responses[2,14,15]. Although past in vitro experiments have clearly shown that elevations in CORT can modify CRH neuron cellular function and excitability[16–18], evidence that these can drive changes in excitability in vivo are lacking. Neurally driven plasticity, which is not stress hormone dependent, has also been demonstrated in vitro[19–21]. Yet, how this type of plasticity shapes in vivo CRH neuron responses is also unclear.

While neurally driven plasticity following stress should be induced quickly, plasticity driven by stress hormone signaling inherently possess a temporal delay in vivo[22]. Thus, neural and hormonal mechanisms would be anticipated to mediate distinct forms of stress-induced adaptation in neural circuits. We therefore set out to understand how stress experience and stress hormone signaling regulate CRH neural activity in vivo. We report that hypothalamic CRH neurons are tonically active in vivo and rapidly respond to threat. Repeated exposure to homotypic stress suppressed threat-evoked CRH neuron activity over a time course of minutes to days. Importantly, this adaptive response did not require stress hormone signaling. While CORT feedback had no effect on the magnitude of threat-evoked activity, it did induce a slow suppression of tonic CRH neural activity. Together, these data reveal that neural and endocrine mechanisms regulate different components of hypothalamic CRH neuron activity dynamics.

## Results

**Optical recordings of CRH neuron activity in vivo.** To gain insight into how neural activity in the hypothalamic CRH neuron population is regulated by stress, we performed GCaMP6s fiber photometry (Fig. 1b) in freely behaving adult male Crh-IRES-cre mice[23]. Dual in vitro loose patch recordings and confocal GCaMP6s imaging showed a high correlation between spiking activity and changes in GCaMP6s fluorescence (Supplementary Fig. 1B–E), confirming that GCaMP6s reliably reports the spiking activity of CRH neurons.

We then characterized the dynamics of CRH neuron population activity in vivo using fiber photometry in resting, stressed, and post-stress conditions (see Supplementary Movie 1 and Fig. 1e, f). In the absence of an external threat stimulus, the CRH neuron population exhibited tonic activity consisting of low irregular GCaMP6s transients (Fig. 1e, f). Under these conditions, circulating CORT levels remained low (samples

taken before ($33.6 \pm 2.4$ ng/mL, $n = 8$; mean $\pm$ SEM) and after ($36.2 \pm 1.8$ ng/mL) 90 min in test apparatus; Fig. 1c). The baseline activity dynamics were not caused by movement or light artifacts as they were not observed in the 405 nm reference channel (Supplementary Fig. 2B) or in control mice expressing GFP (Supplementary Fig. 2C).

Across most experiments, we used a loud white noise (85 dB, 5 min) to induce stress-evoked CRH neuron activation. This type of stressor has been extensively used[24–26] and has the advantage of being highly stereotyped, repeatable, and does not require the experimenter to physically manipulate the animal. CRH neurons were strikingly responsive to this stress stimulus, behaving like a neuronal alarm system as recently reported[9,27]. A strong response was observed almost immediately following the white noise onset, which was followed by a sustained elevation of neural activity during the noise exposure (stress-induced activity). The white noise-induced peak $\Delta F/F$ was $0.89 \pm 0.08$ ($n = 64$; Fig. 1g) and average time to peak from white noise onset was $4.8 \pm 0.32$ s ($n = 64$; Fig. 1h). Mean increase in GCaMP fluorescence during the 5 min white noise exposure was $0.39 \pm 0.03$ $\Delta F/F$ ($n = 64$; Fig. 1i). White noise also robustly increased circulating CORT levels (samples taken 60 min before ($33.5 \pm 2.2$ ng/mL) and 30 min post ($147.5 \pm 10.2$ ng/mL) white noise stress, $n = 8$, $p < 0.001$ paired $t$ test; Fig. 1d).

Interestingly, we observed variability in the CRH neuron activity off-set kinetics after the 5 min white noise (post-stress activity). Some mice exhibited total shut down of activity (return to baseline) almost immediately after the cessation of the white noise (Fig. 1, $F_{1–3}$), while others displayed elevated irregular or sustained activity during the post-stress period (Fig. 1, $E_{1–3}$). When all responses were averaged together, CRH neuron activity returned to baseline levels ($p > 0.05$, repeated-measures (RM) one-way analysis of variance (ANOVA)) at 3.8 min post stress (Fig. 1j). Taken together, these data show that CRH neurons are a vigilant neuronal population that exhibit dynamic responses to novel threat.

**CRH neurons adapt to familiar stress independent of CORT.** We next sought to address how adaptions to stress-evoked CRH neuron responses are tuned by stress hormone feedback. Given the well-described adaptive properties of CORT for learning and stress habituation, we designed a sequential stress protocol to observe the effects of endogenous CORT-negative feedback on CRH activity and stress adaptation. In these experiments, we presented two identical white noise stressors (WN1 and WN2) separated by 30 min ($n = 13$ per group; Fig. 2) or 120 min ($n = 11$ per group; Fig. 3) to mice treated with either vehicle or the CORT synthesis inhibitor, metyrapone.

Following an initial large response to the first white noise (WN1), vehicle-treated mice gradually reduced their CRH neural activity down to basal levels ($n = 13$; Fig. 2a, b). When a second white noise (WN2) was applied following a 30-min interval, both the peak $\Delta F/F$ response (Veh-WN1 $1.0 \pm 0.14$ $\Delta F/F$ vs. Veh-WN2 $0.87 \pm 0.12$ $\Delta F/F$, $p = 0.03$ RM two-way ANOVA; Fig. 2c) and the mean $\Delta F/F$ response (mean CRH activity during 5 min WN: Veh-WN1 $0.42 \pm 0.07$ $\Delta F/F$ vs. Veh-WN2 $0.27 \pm 0.05$ $\Delta F/F$, $p < 0.001$ RM two-way ANOVA; Fig. 2d) were significantly reduced in the vehicle-treated group. We hypothesized that elevated CORT levels consequent from WN1 (Fig. 1d) may mediate the reduced CRH neuron excitability during WN2.

To address the role of CORT feedback in this adaptive change, metyrapone was used to inhibit CORT synthesis (samples taken 60 min before ($35.3 \pm 2.7$ ng/mL) and 30 min post ($39.7 \pm 5.0$ ng/mL) WN, $n = 8$, $p = 0.5$ paired $t$ test; Fig. 2e). Responses to WN1 and the initial post-stress kinetics were virtually identical between the vehicle ($0.42 \pm 0.07$ and $0.15 \pm 0.04$ $\Delta F/F$, 5 min mean

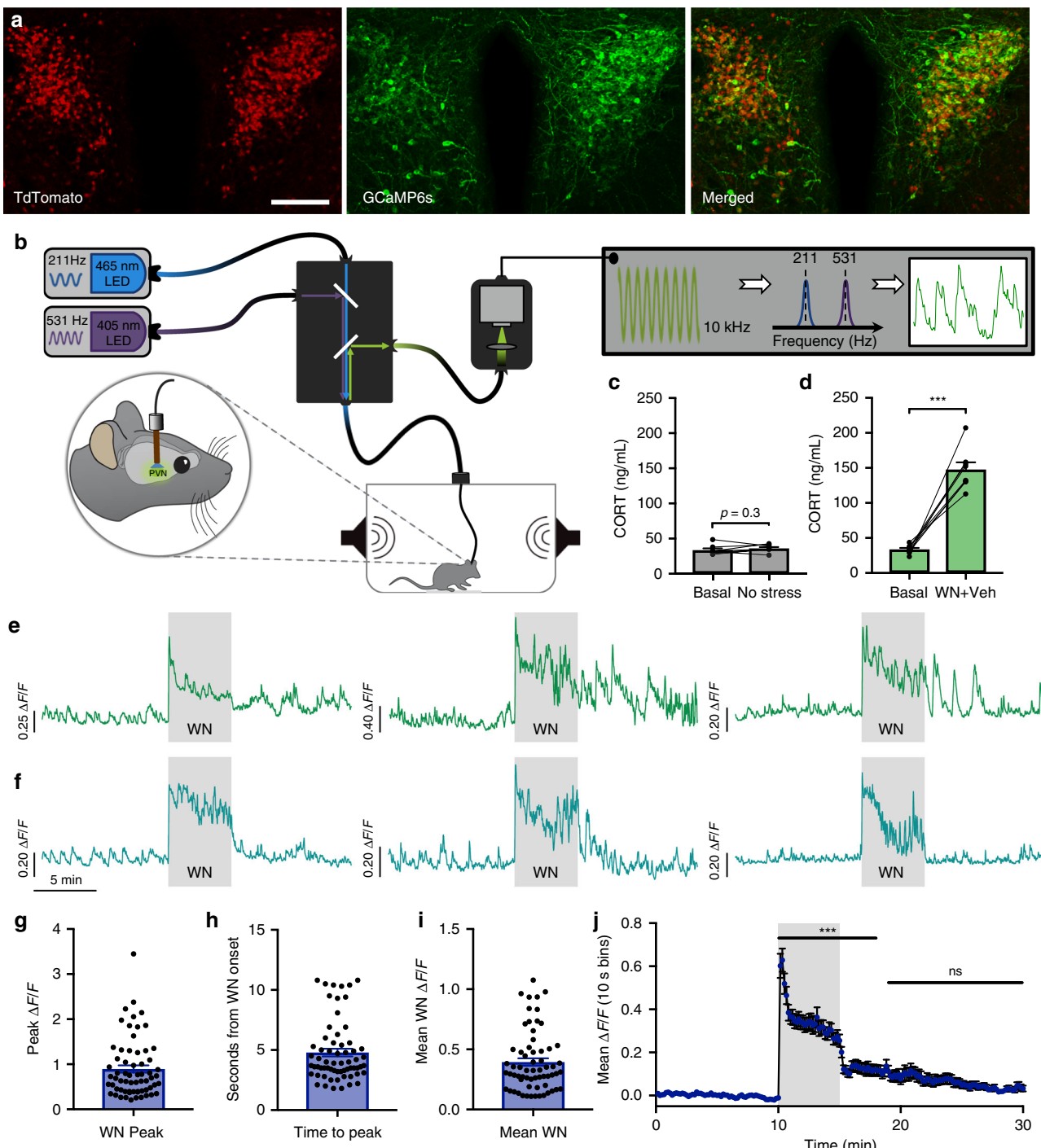

**Fig. 1 Optical recordings of CRH neuron activity in freely behaving mice. a** Image showing PVN expression of CRH-tdTomato reporter (left), AAV-driven GCaMP6s (center), and merged image (right). Scale bar: 100 μm. **b** Schematic illustration of the fiber photometry setup. **c** Blood CORT concentrations obtained from tail tip samples, while mice remained within experimental testing box in the absence of white noise (WN) stress; $n = 8$, $p > 0.05$, paired $t$ test. **d** Blood CORT concentrations following WN stress; $n = 8$, ***$p < 0.001$, paired $t$ test. **e** Photometry recordings of CRH neurons from three individual mice displaying continued activity after termination of WN. **f** Photometry recordings of CRH neurons from three individual mice displaying rapid cessation of activity after termination of WN. **g** Peak $\Delta F/F$ at WN stress onset, $n = 64$. **h** Time to peak from the onset of WN onset, $n = 64$. **i** Mean $\Delta F/F$ of CRH neuron activity during 5 min WN stress from all individual mice tested, $n = 64$. **j** Mean CRH neuron $\Delta F/F$ in 10 sec bins from all individual mice; $n = 64$, repeated-measures (RM) one-way ANOVA ***$p < 0.001$ vs. baseline, Dunn's post hoc test. All data are presented as mean ± SEM.

CRH activity during and after white noise (WN1), respectively; Fig. 2d) and metyrapone-treated groups (0.43 ± 0.07 and 0.19 ± 0.04 $\Delta F/F$, during and after WN1 respectively; Fig. 2d).

Mean CRH neural activity during WN2 was also not different between vehicle and metyrapone treatment groups (mean CRH activity during 5 min WN2: Veh-WN2 0.27 ± 0.05 $\Delta F/F$ vs. MET-WN2 0.33 ± 0.05 $\Delta F/F$, $p = 0.37$ RM two-way ANOVA; Fig. 2d). Furthermore, the mean suppression of WN2 relative to WN1 was not significantly different between treatment groups (Veh-WN2 65.4 ± 6.2% of WN1 response vs. MET-WN2 81.5 ± 8.2% of

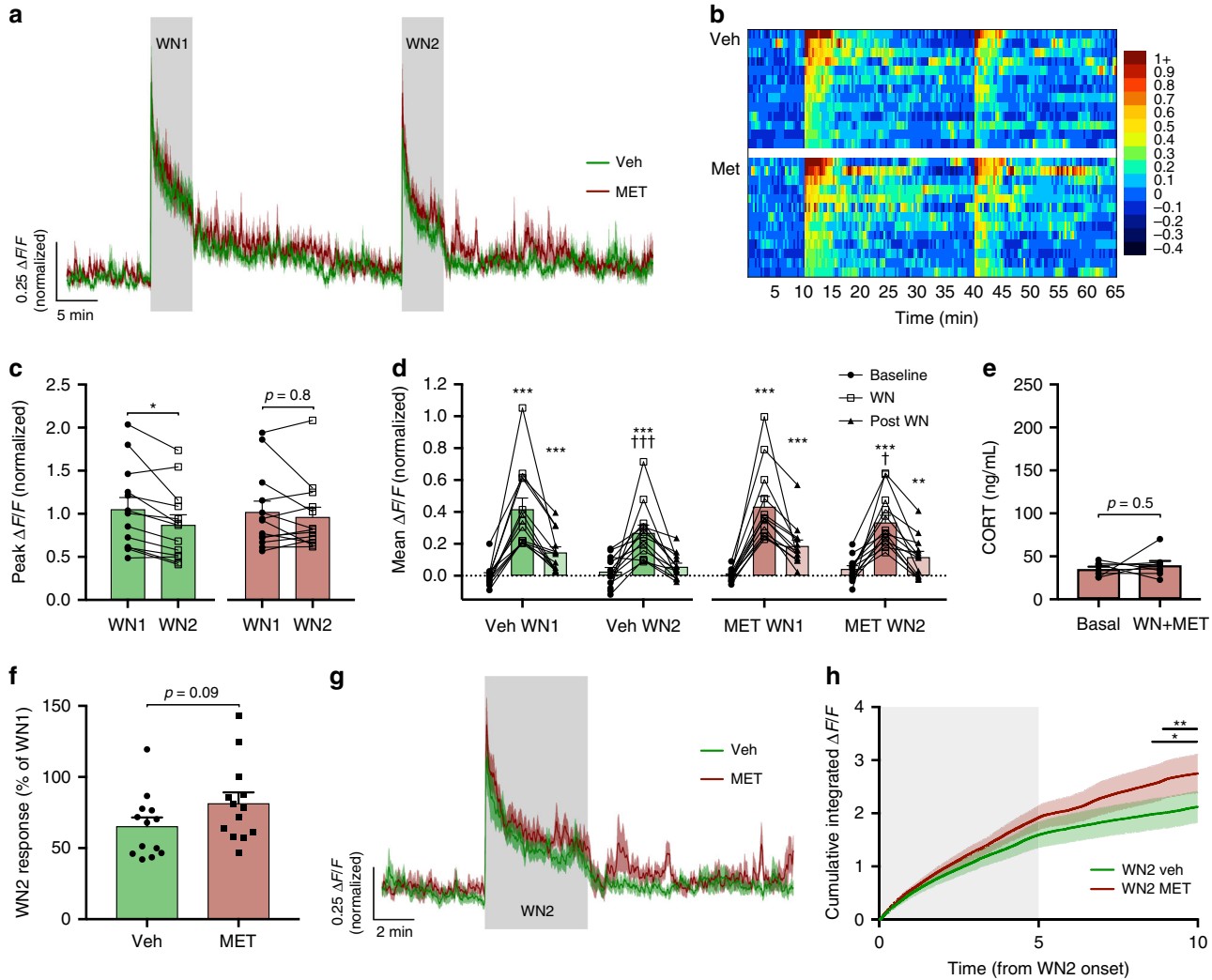

**Fig. 2 CRH neuron shut-off and adaptation occur regardless of fast CORT feedback. a** Mean photometry signals of CRH neuron activity induced by two sequential WN stressors 30 min apart from vehicle and metyrapone-treated mice. **b** Heatmap of mean CRH neuron activity from all individual mice in 20 s bins. **c** Peak $\Delta F/F$ at WN onset; RM two-way ANOVA, *$p < 0.05$ vs. WN1, Holm–Sidak; ANOVA interaction $p = 0.14$, ANOVA main effect of group $p = 0.86$. **d** Average $\Delta F/F$ across 5 min of CRH neuron activity before, during, and after each WN; $n = 13$ per group, RM two-way ANOVA, *$p < 0.05$ vs. baseline, $†p < 0.05$ vs. respective WN1 timepoint, Holm–Sidak; ANOVA interaction $p = 0.90$, ANOVA main effect of group $p = 0.38$. **e** CORT levels before and after WN stress following metyrapone treatment; $n = 8$, paired $t$ test. **f** Percentages of CRH neuron activity during WN2 relative to WN1; Veh vs. MET, Mann–Whitney test. **g** Averaged photometry recordings of CRH neuron activity from all vehicle and metyrapone-treated mice showing the response to WN2. **h** Cumulative integrated $\Delta F/F$ from the time of WN2 onset; RM two-way ANOVA, *$p < 0.05$ Veh vs. MET, Holm–Sidak; ANOVA interaction $p < 0.001$, ANOVA main effect of group $p = 0.09$. Lines indicate points at which statistical significance was reached and its duration. Gray shaded area indicates duration of WN. All data are presented as mean ± SEM, */$†p < 0.05$, **/$††p < 0.01$, ***/$†††p < 0.001$.

WN1, $p = 0.09$ Mann–Whitney test; Fig. 2f). It was evident, however, that metyrapone-treated mice appeared to show a marginally elevated level of activity during and after WN2 (Fig. 2g). We therefore analyzed the cumulative $\Delta F/F$ during both the stress response and post-stress periods to detect changes in neural activity that manifest more slowly over time. Indeed, when the cumulative integrated $\Delta F/F$ was compared between groups, metyrapone-treated mice had a higher level of cumulative activity, which reached significance 3.5 min following the termination of WN2 (Veh vs. MET cumulative $\Delta F/F$, $p < 0.05$ at 8.5 min from WN2 onset, RM two-way ANOVA; Fig. 2h). While no significant differences in mean or cumulative $\Delta F/F$ responses were observed during stress, loss of negative feedback led to slightly elevated activity that became evident in the post-stress period.

These small differences in activity became more apparent when we applied a 120 min inter-stress interval ($n = 11$ per group;

Fig. 3a, b). Vehicle and metyrapone-treated mice again exhibited similar responses to WN1 (Veh $0.36 ± 0.06$ and $0.16 ± 0.05$ $\Delta F/F$, 5 min mean CRH activity during and after WN1, respectively; Fig. 3c; MET $0.38 ± 0.05$ and $0.14 ± 0.02$ $\Delta F/F$, during and after WN1, respectively; Fig. 3c). However, significant differences in tonic activity became discernible 30 min post WN1 stress. Vehicle-treated mice exhibited a near-complete shut-off in CRH neuron activity whereas metyrapone-treated mice failed to display this inhibition (Fig. 3a, b) presumably due to the loss of endogenous negative feedback.

Consistent with this, following WN1 stress, the cumulative integrated fluorescence started to decline in vehicle-treated mice (Fig. 3d). We observed that tonic CRH neuron activity during the white noise interval was either the same or below baseline levels in all vehicle-treated mice. However, in metyrapone-treated mice lacking the ability to synthesize CORT de novo, tonic CRH neural activity during this period was either the same or above baseline

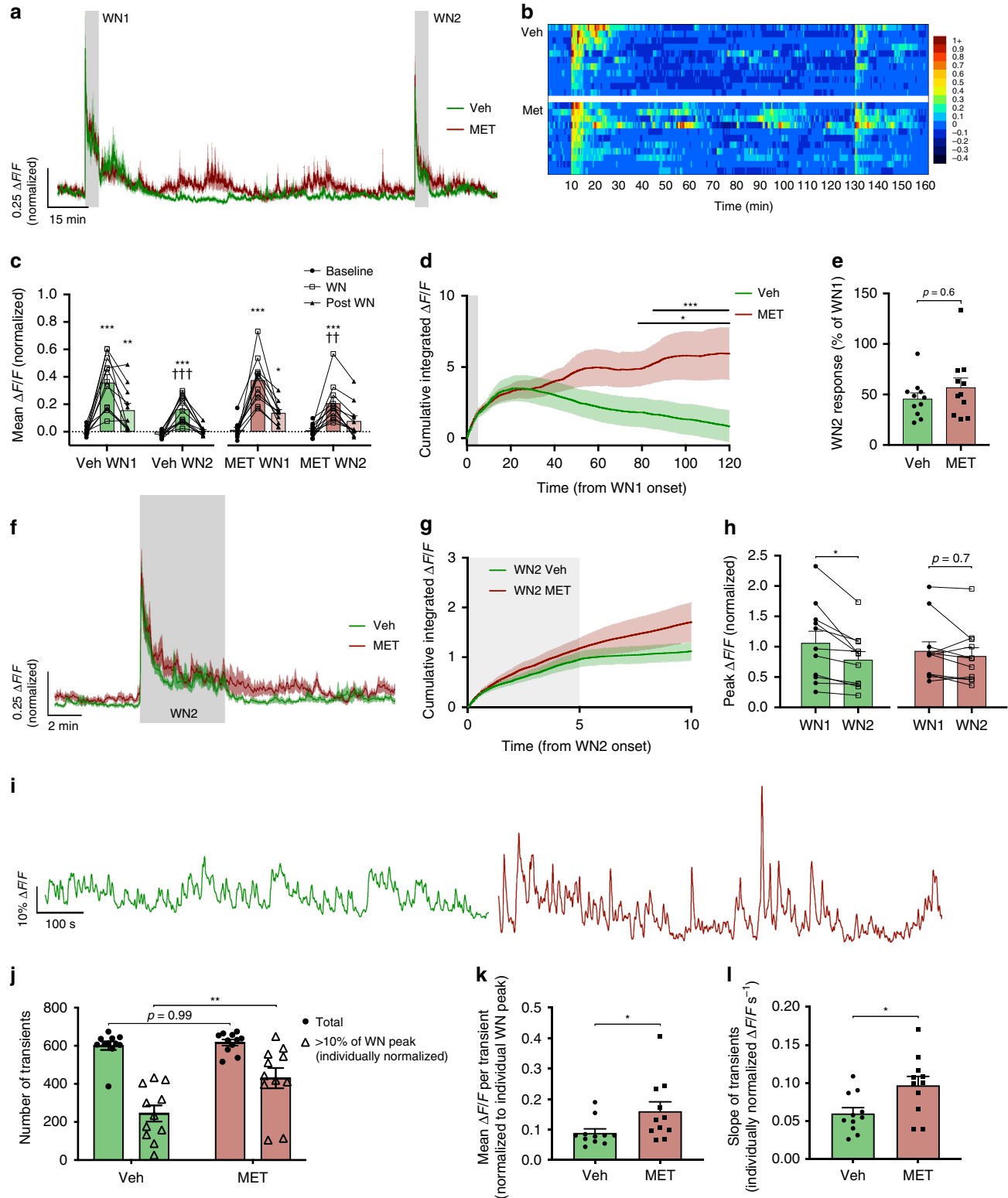

levels. Overall in the metyrapone-treated mice, the mean cumulative CRH neural activity remained elevated (Veh vs. MET cumulative $\Delta F/F$, $p < 0.05$ from 78 min post WN1 onset, RM two-way ANOVA; Fig. 3d).

Despite these differences in tonic activity, a strong adaptive suppression of CRH neuron activity was observed in both groups in response to WN2 (mean CRH activity during 5 min WN2: Veh-WN2 $0.16 \pm 0.03$ $\Delta F/F$ vs. MET-WN2 $0.21 \pm 0.04$ $\Delta F/F$, $p =$

0.98 RM two-way ANOVA; Fig. 3c) and the degree of suppression observed during WN2 was not different between the two conditions (Veh-WN2 $45.9 \pm 5.6\%$ of WN1 response vs. MET-WN2 $57.1 \pm 9.3\%$ of WN1, $p = 0.4$ Mann–Whitney test; Fig. 3e). There was also no significant difference in the cumulative integrated fluorescence response to WN2 between groups ($p = 0.44$ at 10 min from WN2 onset, RM two-way ANOVA; Fig. 3g). Interestingly, while vehicle-treated mice again showed a

**Fig. 3 CORT feedback slowly inhibits tonic CRH neuron excitability without affecting habituation. a** Mean photometry signals of CRH neuron activity from two sequential WN stressors 120 min apart in vehicle and metyrapone-treated mice. **b** Heatmap of mean CRH neuron activity from all individual mice in 30 s bins. **c** Average $\Delta F/F$ across 5 min of CRH neuron activity before, during, and after each WN; $n = 11$ per group, RM two-way ANOVA, *$p < 0.05$ vs. baseline, †$p < 0.05$ vs. respective WN1 timepoint, Holm–Sidak; ANOVA interaction $p = 0.71$, ANOVA main effect of group $p = 0.39$. **d** Cumulative integrated $\Delta F/F$ from the point of WN1 onset; RM two-way ANOVA, *$p < 0.05$ Veh vs. MET, Holm–Sidak; ANOVA interaction $p < 0.001$, ANOVA main effect of group $p = 0.19$. Lines indicate points at which statistical significance was reached and its duration. Gray shaded area indicates duration of WN. **e** CRH neuron activity during WN2 is shown as a percentage relative to WN1; Veh vs. MET, Mann–Whitney test. **f** Averaged photometry recordings of CRH neuron activity from vehicle- and metyrapone-treated mice showing the response to white noise 2 (WN2). **g** Cumulative integrated $\Delta F/F$ from the time of WN2 onset; RM two-way ANOVA, Veh vs. MET, Holm–Sidak; ANOVA interaction $p < 0.001$, ANOVA main effect of group $p = 0.38$. Gray shaded area indicates duration of WN. **h** Peak CRH $\Delta F/F$ at WN onset; RM two-way ANOVA, *$p < 0.05$ vs. WN1, Holm–Sidak; ANOVA interaction $p = 0.08$, ANOVA main effect of group $p = 0.87$. **i** Representative photometry traces of tonic activity (during a period between WN1 and WN2) in individual vehicle- (left) or metyrapone- (right) treated mice. **j** Total number of GCaMP transients and proportion of transients larger than 10% of individual peak WN $\Delta F/F$; *$p < 0.05$, one-way ANOVA, Tukey. **k** Average $\Delta F/F$ (% of individual WN peak) of detected GCaMP transients during post-stress activity; *$p < 0.05$, Mann–Whitney test. **l** Average slope (measured by amplitude/time) of detected GCaMP transients ($\Delta F/F$ measured as % of individual WN peak); *$p < 0.05$, Mann–Whitney test. All data presented as mean ± SEM, */†$p < 0.05$, **/††$p < 0.01$, ***/†††$p < 0.001$.

significant reduction in the peak response to WN2 (WN1 1.0 ± 0.19 $\Delta F/F$ vs. WN2 0.78 ± 0.14 $\Delta F/F$; $p = 0.02$ RM two-way ANOVA; Fig. 3h), this was not observed in metyrapone-treated mice (WN1 0.93 ± 0.15 $\Delta F/F$ vs. WN2 0.84 ± 0.14 $\Delta F/F$; $p = 0.7$ RM two-way ANOVA; Fig. 3h).

Given the significant effects of CORT feedback on tonic CRH activity, we decided to analyze the calcium transients during the inter-stress interval (from 15 min post-WN1, 100 min analysis; Fig. 3i). While the total number of GCaMP6s calcium events were not different between groups ($p = 0.99$ one-way ANOVA; Fig. 3j), metyrapone-treated mice displayed an increased proportion of larger (>10% of individual WN1 peak values) fluorescent transients ($p = 0.005$ one-way ANOVA; Fig. 3j). The overall mean amplitude of transients was also higher in metyrapone-treated mice ($p = 0.03$ Mann–Whitney $t$ test; Fig. 3k). Interestingly, we also observed significantly faster rise times for fluorescent transients in the metyrapone group ($p = 0.03$ Mann–Whitney $t$ test; Fig. 3l). Therefore, the apparent inhibitory effects of CORT feedback are likely caused by reductions in event amplitudes, but not total event frequency, driving an offset in GCaMP6s fluorescence during tonic activity. These differences in tonic calcium events cannot be explained by differences in overall GCaMP fluorescence as there was no significant difference in the peak WN1 response (Fig. 3h) and mean responses to WN1 stress between groups (Fig. 3c).

Despite the significant CORT inhibition of tonic CRH activity, these results indicate that CORT is not involved in adaptive suppression of stress-evoked responses. Instead, past experience alone appeared to be sufficient to induce adaptation. Based on this observation, we theorized that CORT feedback preferentially modulates tonic CRH neuron activity, whereas adaptive changes to stress-evoked CRH neuron drive is experience gated.

**CORT slowly inhibits tonic, but not stress-induced activity.** We next tested whether exogenous CORT could inhibit stress-evoked CRH neural activity in response to a novel stressor. Previous work has consistently shown that exogenous CORT induces a strong suppression in stress-induced endocrine responses[24,28–31], which has often been attributed to inhibition of CRH neuron activity[16,28,31].

All mice were treated with metyrapone 90 min prior to experimental manipulation to block endogenous CORT synthesis. A subsequent single intraperitoneal (i.p.) injection of CORT (11β,21-dihydroxy-4-pregnene-3,20-dione; 0.5 mg/kg in 0.86% dimethyl sulfoxide (DMSO)) rapidly induced high, physiological levels of circulating CORT, whereas vehicle-injected mice showed no change (Supplementary Fig. 3A). Following an initial response to the handling and injection stress, CRH neuron activity rapidly

returned to baseline levels (Fig. 4a, b). We continued to record the tonic CRH neuron activity in the absence of external stress and observed a slow inhibition of tonic activity in the CORT treatment group (−0.08 ± 0.01 $\Delta F/F$ from baseline, 85–125 min post injection, $n = 12$, $p < 0.0001$ RM two-way ANOVA; Fig. 4c). Vehicle-treated controls did not exhibit the same slow inhibition, but instead sustained a stable level of tonic CRH activity (0.004 ± 0.01 $\Delta F/F$ from baseline, 85–125 min post injection, $n = 12$, $p = 0.98$ RM two-way ANOVA; Fig. 4c). The time course of the CORT inhibitory effect was most apparent when the cumulative integrated $\Delta F/F$ was plotted (Fig. 4d). This revealed that while the baseline activity between the vehicle and CORT groups started to diverge ~40 min following injection, a significant difference was not observed until 80 min (Veh vs. CORT cumulative $\Delta F/F$, $p < 0.05$ from 80 min post injection, RM two-way ANOVA; Fig. 4d).

To test the impact of CORT feedback on novel stress-evoked CRH neuron responses, we next applied a white noise stress either 30 min ($n = 12$, Fig. 5a, b) or 150 min ($n = 8$; Fig. 4e) from the time of injection. These time points were chosen based on the kinetics of the inhibition by CORT feedback on tonic CRH neuron activity. The separation of baseline activity is not discernible during the initial 30 min, but persists long after the injection. These time points also correlate to the fast non-genomic and slow genomic time windows of CORT actions[32].

When a novel white noise stress was applied 150 min post injection ($n = 8$; Fig. 4e), CORT treatment seemingly suppressed CRH neuron activity (mean activity during 5 min WN: Veh 0.39 ± 0.08 $\Delta F/F$ vs. CORT 0.23 ± 0.03 $\Delta F/F$, $p = 0.03$ RM two-way ANOVA; Fig. 4f). This inhibition appeared to result from the offset in baseline fluorescence induced by negative feedback rather than a change in the magnitude of the stress response itself. To determine if this was the case, we normalized the baseline to 10 min prior to the white noise to isolate the magnitude of the stress response (Fig. 4h). This revealed negligible differences in the CRH neuron response to white noise between groups (mean activity during 5 min WN: Veh 0.37 ± 0.09 $\Delta F/F$ vs. CORT 0.29 ± 0.04 $\Delta F/F$, $p = 0.6$ RM two-way ANOVA; Fig. 4i).

Responses to a novel white noise stress 30 min post injection ($n = 12$, Fig. 5a, b) were also not different between vehicle and CORT-injected groups (mean activity during 5 min WN; Veh 0.42 ± 0.07 $\Delta F/F$ vs. CORT 0.29 ± 0.05 $\Delta F/F$, $p = 0.16$ RM two-way ANOVA; Fig. 5c). Furthermore, peak response (Veh 1.0 ± 0.14 $\Delta F/F$ vs. CORT 0.9 ± 0.09 $\Delta F/F$, $p = 0.6$ Mann–Whitney test; Fig. 5d) and post white noise activity (Veh 0.11 ± 0.03 $\Delta F/F$ vs. CORT 0.03 ± 0.04 $\Delta F/F$, $p = 0.4$ RM two-way ANOVA; Fig. 5c) were also unaffected by CORT at the 30 min time point. However, when the cumulative integrated $\Delta F/F$ was compared between vehicle and CORT groups, a small suppression of activity in the CORT condition was discernible (Fig. 5e), consistent with our

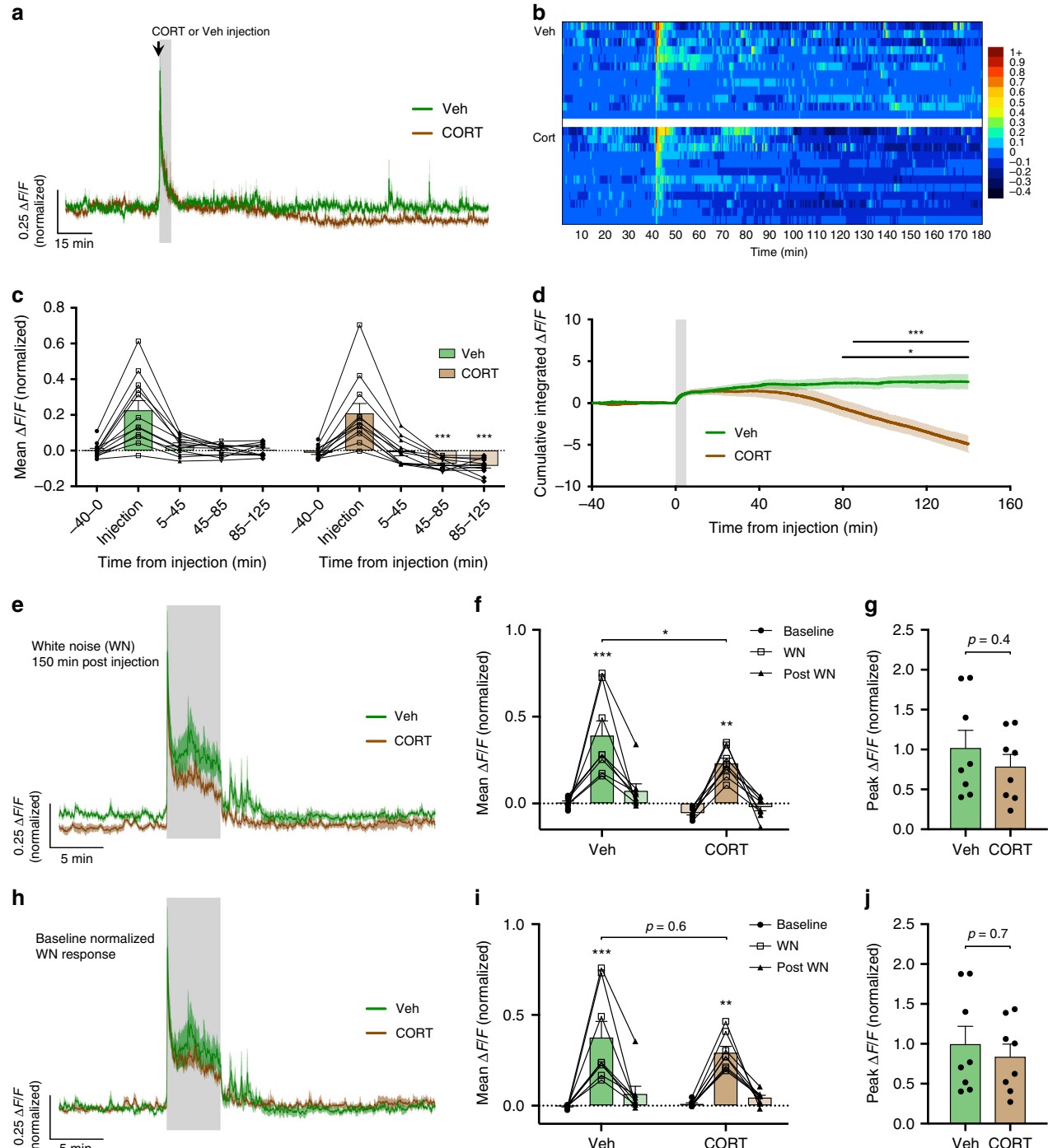

**Fig. 4 CORT-negative feedback slowly suppresses basal but not stress-evoked CRH neuron activity. a** Mean photometry signals of CRH neuron activity from vehicle- and CORT-treated mice. Gray shaded area indicates time of injection where handling/injection stress response is evident. **b** Heatmap of mean CRH neuron activity from all individual mice in 30 s bins. **c** Mean $\Delta F/F$ changes in 40 min bins; $n = 12$ per group, RM two-way ANOVA, ***$p < 0.001$ vs. baseline, Holm–Sidak; ANOVA interaction $p < 0.001$, ANOVA main effect of group $p < 0.001$. CRH neuron activity during injection stress (5 min bin) was not included in the statistical analysis. **d** Cumulative integrated $\Delta F/F$ of tonic CRH neuron activity; RM two-way ANOVA, *$p < 0.05$ Veh vs. CORT, Holm–Sidak; ANOVA interaction $p < 0.001$, ANOVA main effect of group $p = 0.025$. Lines indicate points at which statistical significance was reached and its duration. Gray shaded area indicates stress response due to injection. **e** Averaged photometry recordings of the WN stress response 150 min after injection without baseline normalization. **f** Average $\Delta F/F$ across 5 min of CRH neuron activity before, during, and after WN stress without baseline normalization; $n = 8$ per group, RM two-way ANOVA, *$p < 0.05$ vs. baseline (unless otherwise indicated), Holm–Sidak; ANOVA interaction $p = 0.45$, ANOVA main effect of group $p = 0.01$. **g** Peak $\Delta F/F$ at WN onset without baseline normalization; Mann–Whitney test. **h** Averaged photometry recordings of the WN stress response 150 min after injection with baseline normalized to 10 min of activity prior to white noise. **i** Average $\Delta F/F$ across 5 min of CRH neuron activity before, during, and after WN stress after baseline normalization; $n = 8$ per group, RM two-way ANOVA, *$p < 0.05$ vs. Baseline, Holm–Sidak; ANOVA interaction $p = 0.48$, ANOVA main effect of group $p = 0.47$. **j** Peak $\Delta F/F$ at WN onset after baseline normalization; Mann–Whitney test. All data are presented as mean ± SEM, *$p < 0.05$, **$p < 0.01$, ***$p < 0.001$.

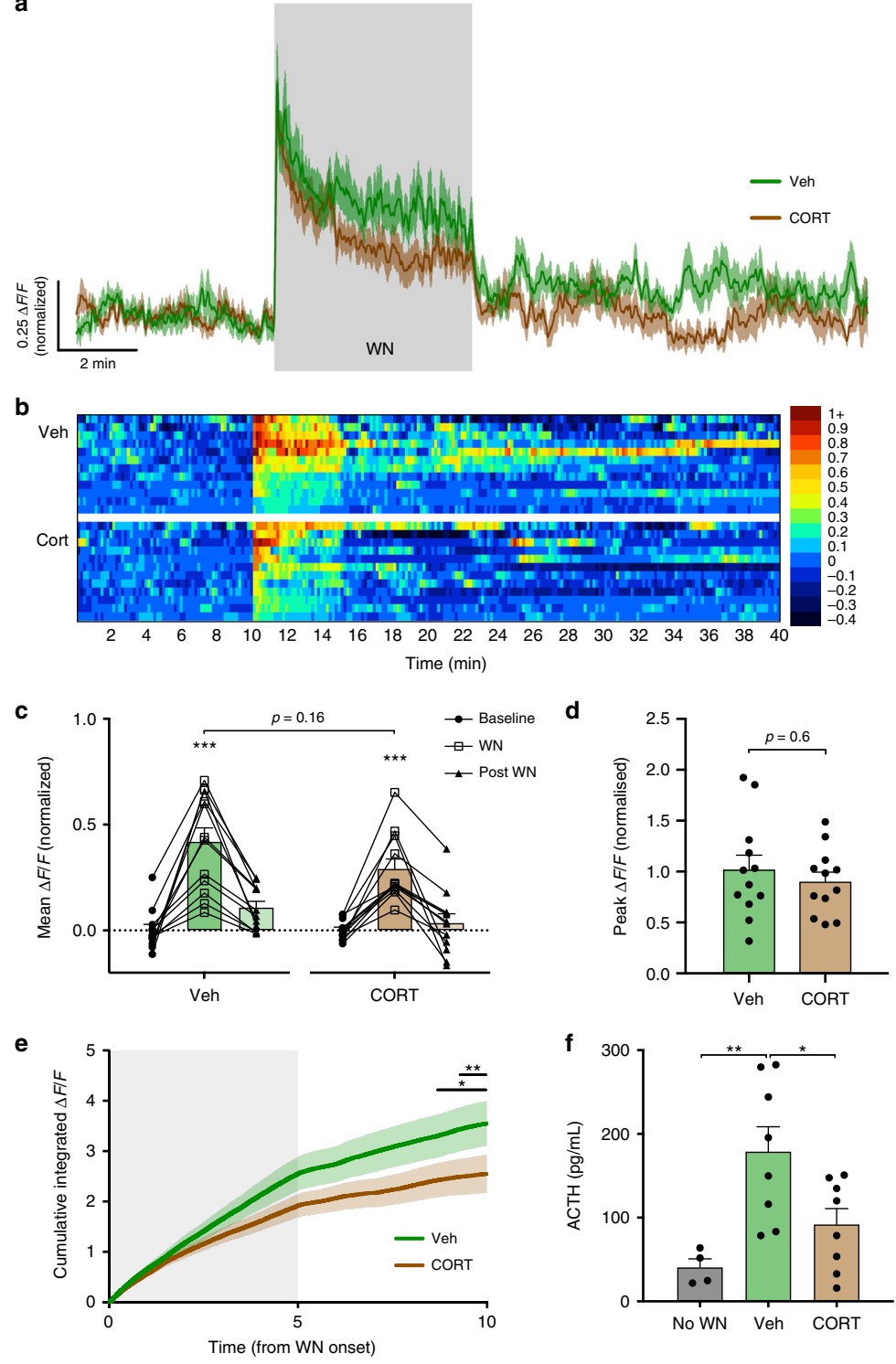

**Fig. 5 Fast CORT feedback has little impact on CRH neural activity but faithfully suppresses ACTH release. a** Mean photometry signals of CRH neuron activity during WN stress, 30 min after injection of either vehicle or CORT. **b** Heatmap of mean CRH neuron activity from all individual mice in 10 s bins. **c** Average ΔF/F across 5 min of CRH neuron activity before, during, and after WN; $n = 12$ per group, RM two-way ANOVA, ***$p < 0.001$ vs. baseline, Holm–Sidak; ANOVA interaction $p = 0.19$, ANOVA main effect of group $p = 0.14$. **d** Peak ΔF/F at WN onset; Mann–Whitney test. **e** Cumulative integrated ΔF/F from the time of WN stress; RM two-way ANOVA, *$p < 0.05$ Veh vs. CORT, Holm–Sidak; ANOVA interaction $p < 0.001$, ANOVA main effect of group $p = 0.21$. Lines indicate points at which statistical significance was reached and its duration. Gray shaded area indicates duration of WN. **f** Plasma ACTH levels 5 min post WN; one-way ANOVA, Tukey's multiple comparison test. All data presented as mean ± SEM, *$p < 0.05$, **$p < 0.01$, ***$p < 0.001$.

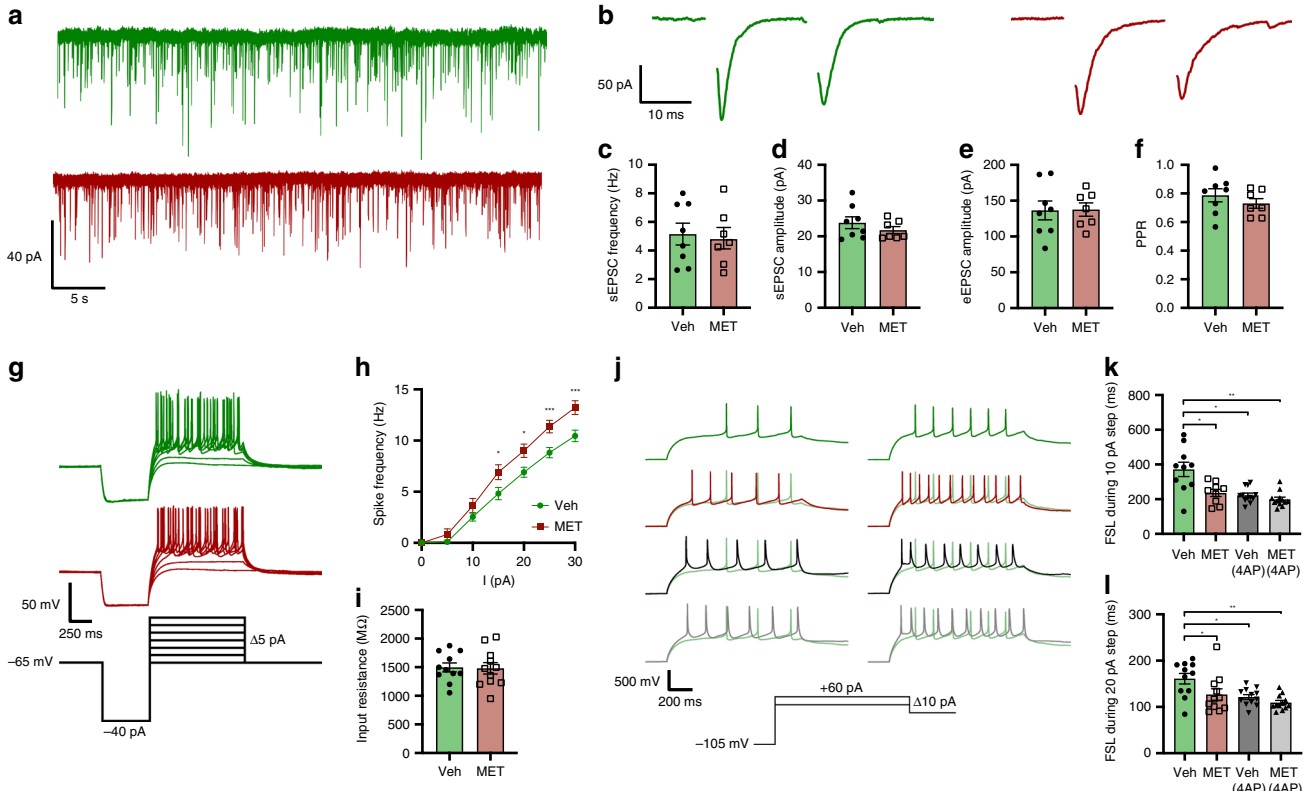

**Fig. 6 CORT feedback increases first spike latency and decreases spike output. a** Traces of sEPSC from individual CRH neurons from vehicle (green) or metyrapone- (red) treated mice. **b** Traces of paired-pulse eEPSC amplitudes from CRH neurons from vehicle (green) or metyrapone- (red) treated mice. **c** Mean sEPSC frequency; unpaired *t* test, $p = 0.8$. **d** Mean sEPSC amplitudes; unpaired *t* test, $p = 0.3$. **e** Mean eEPSC P1 amplitudes; unpaired *t* test, $p = 0.9$. **f** Paired pulse ratio (PPR); Mann–Whitney test, $p = 0.3$. **g** Traces of CRH neuron spike output from vehicle (green) or metyrapone- (red) treated mice in response to varying current steps. **h** Spike frequency plotted for each 5 pA step; $n = 11$ per group, RM two-way ANOVA, *$p < 0.05$ vs. vehicle, Holm–Sidak; ANOVA interaction $p = 0.09$, ANOVA main effect of group $p < 0.001$. **i** Input resistance from individual CRH neurons; unpaired *t* test, $p = 0.9$. **j** CRH neuron responses to 10 pA (left) and 20 pA (right) current steps with or without prior incubation with 4AP (Veh + 4AP in black, MET + 4AP in gray). **k, l** FSL during 10 pA (**k**) and 20 pA (**l**) steps from vehicle and metyrapone-treated mice with or without 4AP incubation. RM two-way ANOVA *$p < 0.05$ Holm–Sidak; ANOVA interaction $p = 0.003$, ANOVA main effect of group $p < 0.001$. All data are presented as mean ± SEM, *$p < 0.05$, **$p < 0.01$, ***$p < 0.001$.

previous observations (Fig. 2h). Specifically, the cumulative integrated $\Delta F/F$ became significantly different between the vehicle and CORT groups 3.5 min following the termination of white noise (Fig. 5e).

We also assessed the effects of exogenous CORT on pituitary adrenocorticotropic hormone (ACTH) secretion. Control ACTH values obtained from metyrapone-treated mice without white noise were $40.6 \pm 10.3$ pg/mL (no WN; Fig. 5f). In response to white noise stress, ACTH levels from vehicle-treated mice were $178.8 \pm 29.7$ pg/mL (measured 5 min after white noise stress). However, in mice previously treated with CORT, stress-evoked ACTH levels were $91.9 \pm 19.0$ pg/mL, which was significantly lower than the vehicle-treated group ($p = 0.04$, one-way ANOVA; Fig. 5f). These results show that fast CORT-negative feedback suppresses ACTH secretion while having a minor impact on CRH neuron activity.

**Cellular correlates of CORT-negative feedback.** The lack of CORT actions on stress-induced CRH neural activity in vivo challenges the long-standing textbook definitions of negative feedback. The most well-characterized fast, non-genomic effect of CORT on CRH neuron excitability is the suppression of spontaneous excitatory postsynaptic current (sEPSC) frequency[16,33]. Surprisingly, while it is generally accepted that CORT reduces CRH neuron spontaneous activity in acute brain slices, there is a

lack of evidence to support an effect of CORT feedback on evoked CRH neuron activity.

Given the suppression of tonic CRH neuron excitability induced by CORT-negative feedback, we sought to identify potential plasticity mechanisms, which may underlie this. Crh-IRES-Cre;Ai14 mice were treated with either vehicle or metyrapone and subsequently exposed to a single white noise stress as described above (Figs. 2 and 3). We then prepared acute brain slices containing the PVN 60 min after white noise stress and analyzed parameters of intrinsic and synaptic excitability using whole-cell patch-clamp electrophysiology.

No differences in sEPSC frequency or amplitude were observed between vehicle- and metyrapone-treated mice following white noise stress (mean sEPSC frequency: Veh $5.15 \pm 0.76$ Hz, $n = 8$ vs. MET $4.86 \pm 0.75$ Hz, $n = 7$, $p = 0.79$ unpaired *t* test; mean sEPSC amplitude: Veh $23.8 \pm 1.64$ pA, $n = 8$ vs. MET $21.7 \pm 1.0$ pA, $p = 0.32$ unpaired *t* test; Fig. 6a, c, d). Likewise, there were also no differences in evoked EPSC (eEPSC) amplitude or paired pulse ratio (PPR) between groups (mean eEPSC amplitude: Veh $136.4 \pm 13.2$ pA, $n = 8$ vs. MET $137.5 \pm 9.3$ pA, $n = 7$, $p = 0.95$ unpaired *t* test; PPR: Veh $0.79 \pm 0.05$, $n = 8$ vs. MET $0.73 \pm 0.03$, $n = 7$, $p = 0.33$ Mann–Whitney test; Fig. 6b, e, f). This lack of effect may be due to the fact that CORT will not remain elevated in brain slices maintained in vitro. Indeed, when CORT (1 μM) was bath applied to brain slices from stress-naive mice, we observed a fast

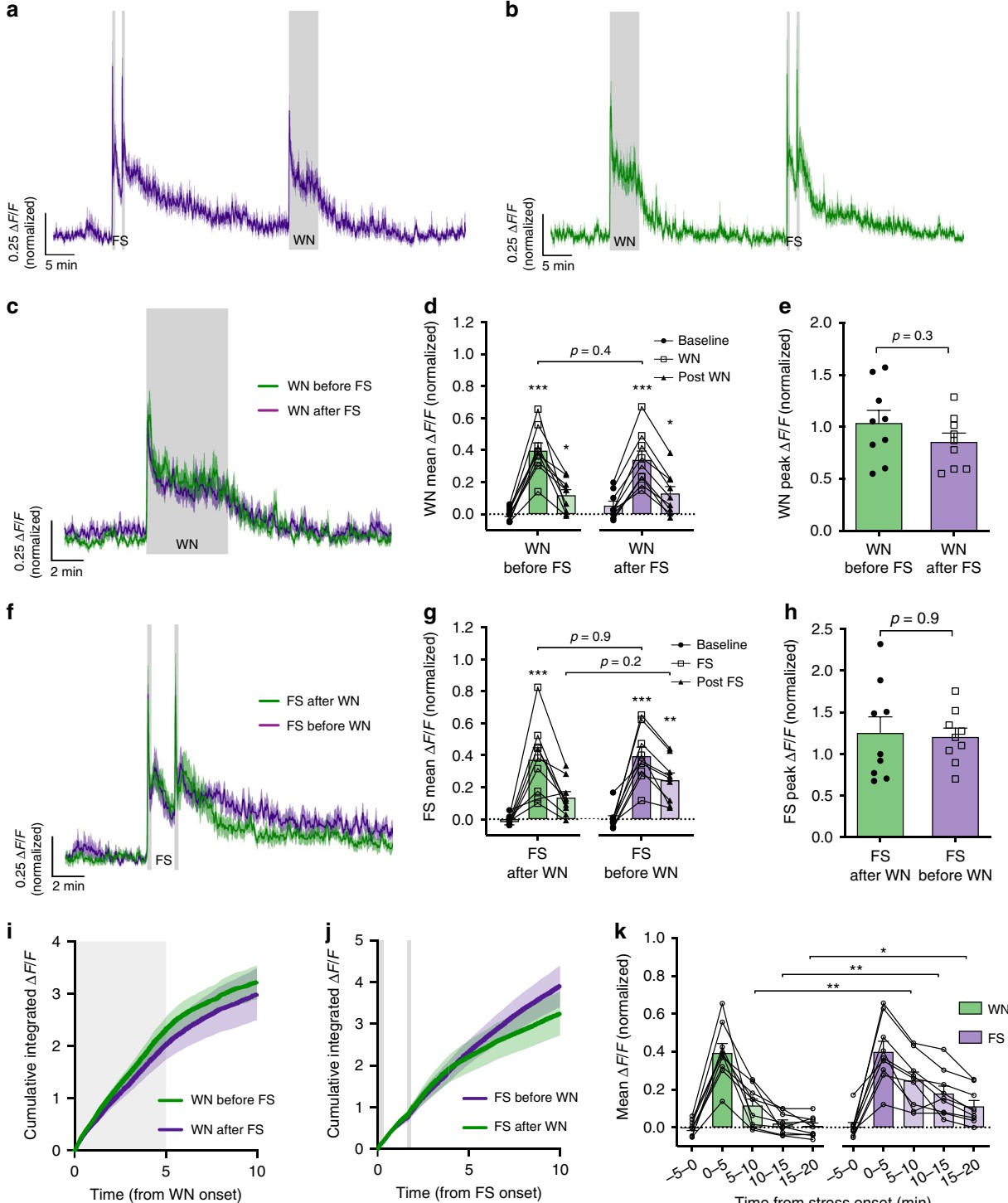

and significant reduction in sEPSC frequency recorded from CRH neurons (Supplementary Fig. 4A, B). However, bath applied CORT had no impact on eEPSC amplitude (Supplementary Fig. 4D, E) or PPR (Supplementary Fig. 4F). Likewise, bath applied CORT also failed to inhibit electrical stimulation induced elevations in excitability in GCaMP6s-expressing CRH neurons (Supplementary Fig. 4G, H).

We next examined whether in vivo CORT-negative feedback would alter the intrinsic excitability of CRH neurons in the presence of AMPA ((+/−)-α-amino-3-hydroxy-5-methylisoxazole-4-propionic acid) and GABA$_A$ (gamma-aminobutyric acid [A]) receptor blockers. To measure intrinsic excitability in

current clamp, CRH neurons were given a family of depolarizing current steps after a single −40 pA hyperpolarizing step (Fig. 6g). We found that CRH neurons from metyrapone-treated mice exposed to WN stress had higher intrinsic excitability and a greater spike output compared to vehicle controls ($p < 0.05$ from 15 pA step onwards, $n = 11$ (Veh) and $n = 12$ (MET), RM two-way ANOVA; Fig. 6h). Input resistance was not different between groups (Veh $1.50 \pm 0.08$ GΩ, $n = 11$ vs. MET $1.48 \pm 0.10$ GΩ, $n = 11$, $p = 0.9$ unpaired $t$ test; Fig. 6i).

Previous work has suggested that CORT-negative feedback enhances a transient outward K$^+$ current in CRH neurons to regulate first spike latency (FSL) and excitability[17]. Consistent

**Fig. 7 Experience-gated habituation to white noise stress is dependent on stress familiarity. a** Mean photometry signals of CRH neuron activity from mice receiving footshock (FS) followed by WN with a 30 min interval. **b** Mean photometry signals from mice receiving WN followed by FS. **c** Averaged photometry recordings of CRH neuron activity during WN stress when presented either first or second. **d** Average $\Delta F/F$ across 5 min of CRH neuron activity before, during, and after each WN; $n = 9$ per group, RM two-way ANOVA, $*p < 0.05$ vs. baseline, Holm–Sidak; ANOVA interaction $p = 0.13$, ANOVA main effect of group $p = 0.96$. **e** Peak $\Delta F/F$ response to WN stress presented either first or second; Mann–Whitney test. **f** Averaged photometry recordings of CRH neuron activity during footshock stress when presented either first or second. **g** Average $\Delta F/F$ across 5 min of CRH neuron activity before, during, and after each FS; $n = 9$ per group, RM two-way ANOVA, $*p < 0.05$ vs. baseline, Holm–Sidak; ANOVA interaction $p = 0.44$, ANOVA main effect of group $p = 0.31$. **h** Peak $\Delta F/F$ response to FS stress presented either first or second; Mann–Whitney test. **i** Cumulative integrated $\Delta F/F$ from the time of WN stress; RM two-way ANOVA, $p > 0.99$ WN before FS vs. WN after FS at 10 min, Holm–Sidak; ANOVA interaction $p > 0.99$, ANOVA main effect of group $p = 0.52$. Gray shaded area indicates duration of WN. **j** Cumulative integrated $\Delta F/F$ from the time of FS stress; RM two-way ANOVA, $p > 0.99$ FS before WN vs. FS after WN at 10 min, Holm–Sidak; ANOVA interaction $p < 0.001$, ANOVA main effect of group $p = 0.67$. Gray shaded areas indicate timing of the two FSs (2 s duration each). **k** Mean $\Delta F/F$ response to the first presentations of WN or FS stress in 5 min bins; $n = 9$ per group, RM two-way ANOVA, $*p < 0.05$ WN vs. FS, Holm–Sidak; ANOVA interaction $p = 0.007$, ANOVA main effect of group $p = 0.051$. All data presented as mean ± SEM, $*p < 0.05$, $**p < 0.01$, $***p < 0.001$.

with this mechanism, CRH neurons from metyrapone-treated mice had a significantly shorter FSL compared to vehicle controls (FSL at 10 pA step: Veh 371.5 ± 41.4 ms vs. MET 235.6 ± 20.6 ms, $p = 0.02$ RM two-way ANOVA, $n = 10$ (Veh) and $n = 9$ (MET); Fig. 6k. FSL at 20 pA step: Veh 161 ± 11.4 ms vs. MET 126.8 ± 12.4 ms, $p = 0.03$ RM two-way ANOVA, $n = 11$ (Veh) and $n = 11$ (MET); Fig. 6l). To determine whether differences in FSL induced by negative feedback were mediated by an outward $K^+$ conductance, we used 4-aminopyridine (4AP, 2 mM) to inhibit rapidly inactivating $K^+$ currents. FSL delays in CRH neurons from vehicle-treated mice following 4AP incubations were comparable to that of metyrapone-treated mice (FSL at 10 pA step: Veh + 4AP 222.5 ± 12.62 ms vs. MET 235.6 ± 20.6 ms, $p = 0.48$ RM two-way ANOVA, $n = 12$ (Veh + 4AP) and $n = 11$ (MET); Fig. 6k. FSL at 20 pA step: Veh + 4AP 121.4 ± 5.3 ms vs. MET 126.8 ± 12.4 ms, $p = 0.68$ RM two-way ANOVA, $n = 12$ (VEH + 4AP) and $n = 11$ (MET); Fig. 6l). Whereas 4AP had no significant effects on the FSL of CRH neurons from metyrapone-treated mice (Fig. 6k, l).

Overall, these data show that following stress, CORT-negative feedback suppresses intrinsic excitability and prolongs FSL. The timing of suppressed intrinsic excitability following stress matches the timing of inhibition of tonic CRH neuron activity measured with fiber photometry in vivo. Second, the faster and larger spontaneous calcium events observed in the metyrapone group in vivo (Fig. 3j–l) are consistent with higher spiking excitability and shorter delay to spike observed in the metyrapone group in vitro.

**Stress familiarity determines adaptation of CRH neuron output.** Regardless of CORT feedback, adaptive stress habituation was consistently observed across the two sequential white noise epochs (Figs. 2 and 3). We proposed that this adaptation is experience gated, and therefore dependent on the familiarity of the stressor itself, but also requires regular exposure to the stress stimulus. To determine whether such adaptation was dependent on stress familiarity, we used a sequential stress paradigm with two different types of stressors: white noise and footshock (two shocks, 0.3 mA, 2 s duration, 100 s interval) separated by 30 min. Two groups of mice received white noise and footshock stressors in alternating order ($n = 9$; Fig. 7a, b). We speculated that adaptive habituation would not be observed with this paradigm due to the differing nature of the stressors. If this were the case, the white noise or footshock responses should be the same magnitude regardless of whether they were presented first or second.

CRH neural responses to white noise stress were near identical regardless of whether the white noise was presented first or 30 min following footshock (mean CRH activity during 5 min WN: before FS, 0.40 ± 0.05 $\Delta F/F$ vs. after FS, 0.34 ± 0.06 $\Delta F/F$, $p = 0.42$ RM two-way ANOVA; Fig. 7c, d). Footshock responses were also

comparable between groups that either received it before or after white noise (mean CRH activity during 5 min from first FS onset: before WN, 0.40 ± 0.06 $\Delta F/F$ vs. after WN, 0.37 ± 0.08 $\Delta F/F$, $p = 0.91$ RM two-way ANOVA; Fig. 7f–g). Peak responses to white noise or footshock (Fig. 7e, h), cumulative $\Delta F/F$ changes during and after white noise or footshock (Fig. 7i, j) were also unaffected by stress order.

To investigate the requirement for regular exposure to the stressor in maintaining long-term adaptation, mice received single daily exposure to white noise stress over 4 consecutive days (D1–4, Fig. 8a). Each exposure promoted adaptive habituation and subsequently suppressed the white noise response the next day. By day 4, white noise elicited merely a startle response, which rapidly returned to baseline levels (mean CRH activity during 5 min WN: day 1 0.47 ± 0.08 $\Delta F/F$ vs. day 2 0.30 ± 0.04 $\Delta F/F$, $p = 0.04$; day 1 vs. day 3 0.22 ± 0.06 $\Delta F/F$, $p = 0.002$; day 1 vs. day 4 0.16 ± 0.06 $\Delta F/F$, $p < 0.001$, RM two-way ANOVA; Fig 8c, d). While mean CRH neural activity during each white noise was consequently diminished (Fig. 8d), peak responses were unchanged, presumably due to a non-adaptive startle response (CRH peak response to WN day 1 1.0 ± 0.14 $\Delta F/F$ vs. day 4 0.78 ± 0.15 $\Delta F/F$, $p = 0.27$, RM two-way ANOVA; Fig. 8e).

This adaptive habituation to white noise stress could be extinguished following 3 weeks of white noise abstinence (round 2 day 1 mean CRH response during 5 min WN: 0.36 ± 0.05 $\Delta F/F$ vs. round 1 day 1, $p = 0.32$, RM two-way ANOVA; Fig. 8b, d), but could be relearnt, demonstrating the requirement for regular stress exposure to maintain the adaptive change. Blood CORT levels 30 min after each white noise significantly correlated with mean $\Delta F/F$ responses across the two 4-day challenges (Fig. 8f; $r = 0.9$, $p = 0.003$, Pearson's correlation coefficient). Together, these data suggest that habituation of CRH neural activity is an important mechanism shaping long-term adaptation of the neuroendocrine stress response.

## Discussion

There is consensus that CORT-negative feedback is essential for inhibiting the CRH neuron stress response. Despite the large body of evidence demonstrating fast and delayed forms of negative feedback on CRH neuron excitability in vitro[16,33,34], how such mechanisms tune CRH neuron activity in vivo has until now been unknown. Using fiber photometry to observe the CRH neuron population activity in awake behaving mice, our findings reveal how this neural population responds to stress and the real-time dynamics of CORT-negative feedback. Specifically, we show that CRH neurons are tonically active in unstressed states and respond rapidly to sensory detection of external threats. We show that removal of the stressor alone is sufficient to initiate a fast decline in CRH activity, independent of CORT synthesis. This finding challenges the idea that CORT negative

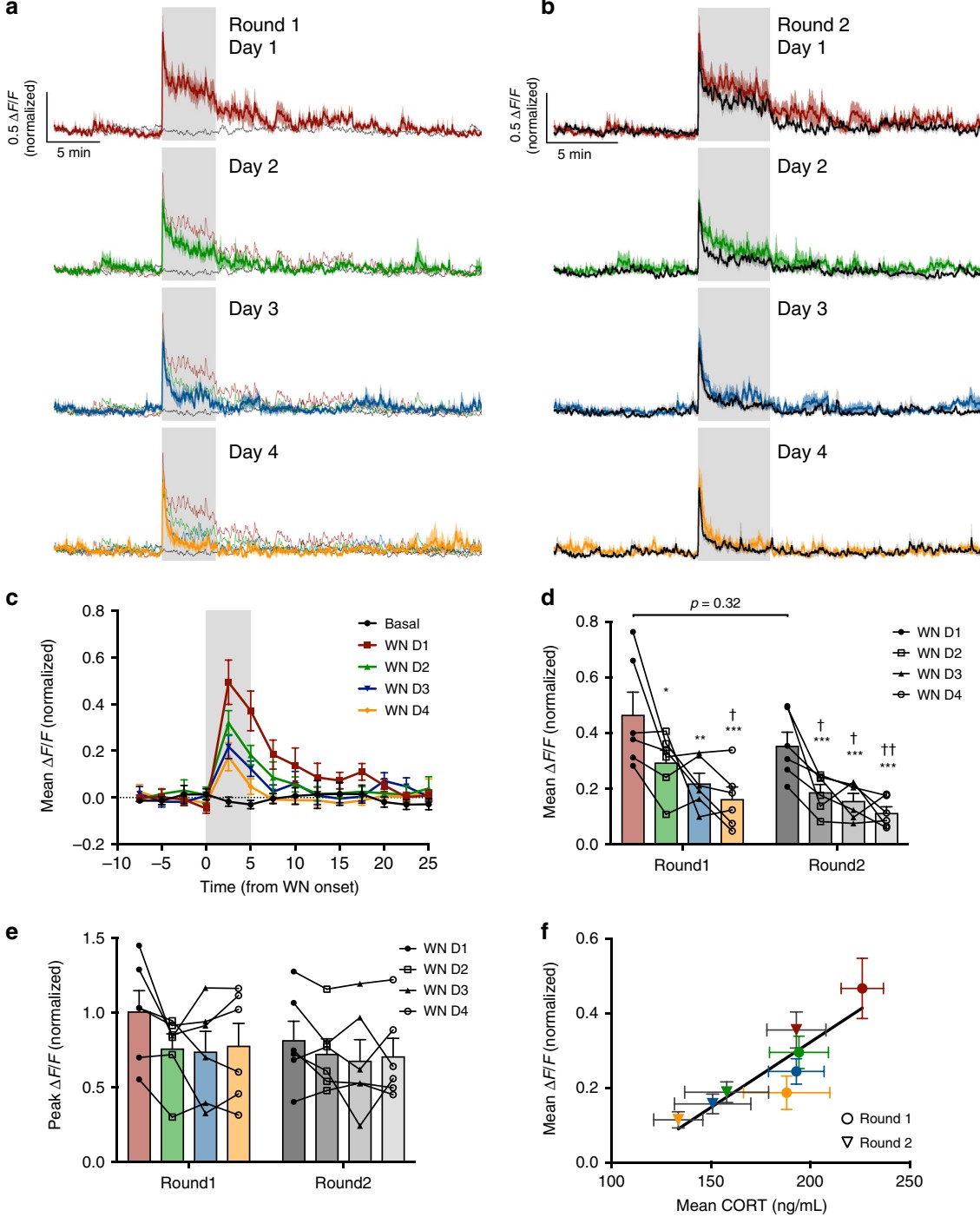

**Fig. 8 Long-term adaptation of CRH neural activity to stress. a** Mean photometry signals of CRH neuron activity from mice receiving daily WN stress over 4 days (round 1: day 1; red; day 2, green; day 3, blue; day 4, orange). Black trace indicates mean CRH neuron activity from the same cohort of mice on day 0, in the absence of WN. **b** Mean photometry signals of CRH neuron activity from the same mice after a 3-week rest interval (with no stress) and then subsequently receiving daily WN stress over 4 more days (round 2: all recordings in black overlaid with corresponding round 1 WN response). **c** Mean $\Delta F/F$ changes in response to each WN or no stress (black) in 2.5 min bins during round 1. **d** Average $\Delta F/F$ of CRH neuron activity during WN stress across the two rounds of repeated stress; $n = 6$, RM two-way ANOVA, $*p < 0.05$ vs. round 1 WN day 1, $\dagger p < 0.05$ vs. round 2 WN day 1, Holm–Sidak; ANOVA interaction $p = 0.77$, ANOVA main effect of group $p = 0.09$. **e** Peak $\Delta F/F$ at WN onset across the two rounds of repeated stress; RM two-way ANOVA, Holm-Sidak; ANOVA interaction $p = 0.53$, ANOVA main effect of group $p = 0.60$. **f** Correlation of mean CRH neuron activity during WN and corresponding post-stress blood CORT concentration across the two rounds (round 1 in circles and round 2 in triangles, 4 days of WN indicated by corresponding color; Pearson's $r = 0.90$, $r^2 = 0.8$. All data presented as mean ± SEM, $*/\dagger p < 0.05$, $**/\dagger\dagger p < 0.01$, $***/\dagger\dagger\dagger p < 0.001$.

feedback induces a fast "shut off" of the neural stress response. While negative feedback is indeed essential for completely returning CRH neuron activity to the baseline state, this is a slow, gradual effect.

In both our exogenous and endogenous negative feedback models, we did not observe any substantial effect of CORT on stress-evoked CRH neural activity. Instead, experience-dependent habituation induced strong suppression of CRH neural responses to stress; whether the sequential white noise stressors were 30 min, 120 min, or 24 h apart. These data show that CRH neuron responses are highly adaptive following repeated homotypic stress. This adaptation would act to prevent excessive stress responses against learnt non-harmful threats to limit unnecessary energy expenditure and maximize survival. Together, these findings reveal the importance of experience dependent plasticity in shaping neural responses to stress and redefine the principles of CORT negative feedback in the stress axis.

Past studies on CORT feedback have injected glucocorticoid receptor agonists and/or antagonists peripherally and observed suppression and enhancement in stress hormone secretion, respectively[14,24,28,29,31]. Our current work shows that fast CORT feedback suppresses stress-evoked ACTH release 30 min following injection. However, we observed only a subtle inhibition of stress-evoked CRH neuron activity during this period (Fig. 5). While we did not observe any changes in basal CRH neuron activity in the first 30 min following CORT injection (Fig. 4), we cannot rule out the possibility that CORT suppressed basal ACTH release during this period. Previous work has clearly shown that CORT negative feedback at the pituitary is important for the overall suppression of HPA axis output. In vitro studies in pituitary corticotroph cells have demonstrated that CORT can suppress electrical excitability, calcium elevations, and ACTH secretion within minutes[35–37]. CORT can also rapidly blunt CRH-stimulated ACTH secretions in vivo[38–40]. While our results challenge long held beliefs regarding CORT actions on stress circuits in the brain, they are in fact consistent with past findings where exogenously applied glucocorticoids fail to impact c-fos expression in the PVN following stress[28,41].

While CORT negative feedback did not substantially change stress-evoked CRH neuron responses, it was able to suppress tonic CRH neuron activity. This mechanism would act to reduce ongoing CRH secretion over extended time frames. We examined the cellular mechanisms which could underlie this CORT-induced, slow suppression of tonic activity. While fast CORT feedback is well known to inhibit spontaneous glutamatergic transmission[16], we found no difference in spontaneous glutamate release following stress in the presence or absence of CORT negative feedback. Instead, stress induced CORT led to changes in intrinsic excitability. Negative feedback prolonged FSL, an effect that could be reversed with the potassium channel blocker 4AP. These findings are consistent with previous work showing that stress-induced CORT elevations reduce CRH neuron intrinsic excitability without affecting glutamate transmission[17]. These data also suggest that regulation of intrinsic excitability may be more important than inhibition of spontaneous glutamate transmission for the CORT induced suppression of tonic CRH neuron activity in vivo.

Our work has also identified that stress familiarity alone, independent of CORT negative feedback, is sufficient to strongly inhibit stress-evoked CRH neuron responses. Importantly, adaptive CRH neuron stress responses were not observed with unfamiliar, heterotypic stressors. Adaptive responses to white noise stress have previously been observed where CORT release is blunted following the second presentation of the same stressor[26]. Our observations suggest that habituation of peripheral stress hormone responses following homotypic stress are mediated by adaptation of CRH neural responses.

While the mean CRH neuron activity during white noise strongly habituated with repeated presentation of the familiar stress over 4 days, the peak response did not habituate. We speculate that the initial fast activation of CRH neurons is due to a startle response, which is important for quickly activating CRH neurons when exposed to an unexpected potential threat. Subsequently, as the animal has time to determine the nature of the threat, the stress response can then be adjusted to the level of danger. In the case of white noise, the animal learns over repeated presentations that the stimulus is non-harmful and therefore CRH neural activity returns towards baseline levels. This processing of "danger" information is likely mediated by upstream neural populations that are synaptically connected to CRH neurons. While the neural circuits involved in stress-specific habituation remains poorly understood, our work reinforces the importance of synaptic regulation of stress-evoked CRH neuron activity in addition to hormonal regulation.

While it may seem counter-intuitive, the overall lack of CORT inhibition of stress-evoked CRH neuron activity may be important for survival. Hypothalamic CRH neurons have recently been shown to serve critical roles in stress-induced behavior[7], pheromone release[8], and encoding of valence[9]. Regardless of CORT milieu, appropriate behavioral responses in dangerous situations remain essential. Therefore, we argue that the lack of CORT effects on stress-evoked CRH neuronal activity may serve an important role in permitting normal neural responses and corresponding CRH-mediated stress behaviors, which facilitate survival.

In summary, we provide novel insight into how CRH neurons respond to stress in freely behaving mice. We have directly addressed the role of CORT-negative feedback on CRH activity for the first time and our results should prompt a reevaluation of the existing textbook definitions of negative feedback. Furthermore, we report that CRH neurons respond rapidly to sensory detection of threat and tune their output depending on stress familiarity. Thus, neural and endocrine mechanisms regulate different aspects of HPA axis function to shape an organism's responses to stress.

## Methods

**Animals**. All mice were housed under a 12 h light/dark cycle in individually ventilated cages with ad libitum access to food and water. All experiments were conducted in accordance with the New Zealand Animal Welfare Act and approved by the University of Otago Animal Welfare and Ethics Committee.

**Stereotaxic surgery**. Adult (10–12-week-old) male Crh-IRES-Cre[23] or Crh-IRES-Cre;Ai14 (tdTomato reporter) mice[42] were anesthetized with 2% isofluorane and placed in a stereotaxic frame. Adeno-associated virus (AAV) encoding GCaMP6s (AAV1.CAG.Flex.GCaMP6s.WPRZ.Sv40) or GFP (AAV9.Syn.DIO.EGFP.WPRE. hGH) was stereotaxically injected unilaterally into the PVN via a Hamilton syringe (−0.8 mm AP, −0.25 mm ML, −4.5 mm DV) at a volume of 1 μL over 10 min. A fiberoptic cannula (400 μm core, 0.48 NA; Doric Lenses) was then implanted at the same coordinates and secured using adhesive dental cement. All mice were given carprofen (5 mg/kg) and lidocaine (2%) during surgery and allowed to recover for 4 weeks before experimental recordings.

**Fiber photometry**. Optical recordings of GCaMP6s fluorescence were acquired using a custom software acquisition system with optical components purchased from Doric Lenses[43]. Excitation LEDs (465 nm blue and 405 nm violet) were sinusoidally modulated at 211 and 531 Hz, respectively. Excitation wavelengths were relayed through a filtered fluorescence minicube (spectral bandwidth: 460–490 and 405 nm) to a 400 μm 0.48 NA fiberoptic cable connected to the mouse. Light power for the 465 nm wavelength at the fiber tip was 35 μW (70 μW/mm²) and was estimated to drop off to 19 μW/mm² within a distance of 0.2 mm from the fiber tip in brain tissue (61% power attenuation). A single emission (filtered at 500–550 nm) was detected using a femtowatt photoreceiver (2151, Newport) with a lensed fiber cable adapter. All signals were acquired at 10 kHz, digitized with a demodulation bandwidth of ±5 Hz, and down-sampled to a rate of 10 Hz.

Due to the duration of our recordings, a linear regression was used to correct for bleaching of the signal using the slope of the 405 nm signal fitted against the 465 nm signal, where $\Delta F/F = (465\ nm - \text{fitted}405)/\text{fitted}405$. We then normalized the $\Delta F/F$ for each experiment using the mean peak response to white noise onset from the vehicle control groups (Veh mean white noise peak = 1.0 normalized $\Delta F/F$). Mice with peak response signals below 25% $\Delta F/F$ were excluded from the study.

All experiments were conducted between zeitgeber time 0–5 in the animal's home cage, which was placed in a custom-made apparatus (40 cm length, 40 cm width, 40 cm height) with white walls and transparent lid. Speakers were mounted to the walls on two sides and a shock grid floor (Kinder Scientific) could be incorporated for induction of loud white noise (85 dB) or footshock (0.3 mA) stress without experimental handling. For experiments involving footshock, a custom bottomless cage was used in place of the home cage. Mice were habituated to the testing room and apparatus for 7 consecutive days prior to experimental manipulations.

CORT injection and white noise experiments: Mice were given an i.p. injection of metyrapone (100 μL bolus i.p.; 75 mg/kg; 25% PEG in saline) 90 min prior to the experiment. A 40-min baseline recording was taken prior to injection of CORT (100 μL bolus i.p.; 0.5 mg/kg; 0.84% DMSO in saline) or vehicle. This dose was chosen from a prior characterization experiment using wild-type C56BL6 mice where repeated tail blood samples were obtained (Supplementary Fig. 3A). This injection dose caused high but physiological blood concentrations of CORT comparable to an acute restraint stress response (Supplementary Fig. 3B). We have also previously observed such levels of CORT[44] and other studies have also reported similar elevations in CORT levels following an acute restraint stress in mice, using the same enzyme-linked immunosorbent assay (ELISA)[45,46]. Two separate groups of mice were used to test white noise responses at either 30 or 150 min post injection. For each experiment, mice were randomly assigned to the vehicle or CORT group first and then received the alternative treatment 4 weeks later. Thus, each mouse served as its own internal control.

Sequential white noise experiments: All mice were given an i.p. injection of either metyrapone (100 μL bolus i.p., 75 mg/kg; 25% PEG in saline) or saline 90 min prior to the experiment. A 10 min baseline recording was taken prior to the onset of the first white noise. Two separate groups of mice were used to test two sequential white noise stress responses at either 30 or 120 min intervals. Each mouse served as their own internal control (metyrapone vs. vehicle) and repeated the experiment 4 weeks later, receiving the alternative treatment. For experiments involving daily white noise stressors, a single cohort of mice received four daily white noise stressors and repeated the daily stress protocol 3 weeks later.

White noise and footshock variable stress: Mice were presented with a footshock (two shocks separated by a 100-s interval, 0.3 mA, 2 s duration) and white noise stress in alternating order with a 30-min interval between each stressor. Each mouse served as their own internal control (footshock then white noise vs. white noise then footshock) and repeated the experiment 4 weeks later, receiving the alternative stress order.

**Brain slice electrophysiology and calcium imaging**. Mice were euthanized via cervical dislocation and brains were sliced in an ice-cold cutting solution containing (in mM): 87 NaCl, 2.5 KCl, 25 NaHCO₃, 1.25 NaH₂PO₄, 0.5 CaCl₂, 6 MgCl₂, 25 D(+)-glucose and 75 sucrose, saturated with 95% O₂/5% CO₂. Acute brain slices (200 μm) containing the PVN were allowed to recover for at least 1 h in artificial cerebrospinal fluid (aCSF) consisting of (in mM): 126 NaCl, 2.5 KCl, 26 NaHCO₃, 1.25 NaH₂PO₄, 2.5 CaCl₂, 1.5 MgCl₂ and 10 D(+)-glucose, saturated with 95% O₂/5% CO₂ at 30–32 °C. All recordings were performed under an Olympus FV1000 confocal microscope at 30 °C with a perfusion rate of 1–2 ml/min. CRH neurons were visualized by either tdTomato or GCaMP6s expression.

For voltage clamp loose patch recordings, borosilicate glass pipettes (~4 MΩ) were filled with aCSF and a low resistance seal (~10 MΩ) was achieved. Noradrenaline (50 μM) or KCl (7.5 mM) was applied to induce action potential firing and changes in GCaMP6s fluorescence were simultaneously imaged using a 488 nm Argon laser (Melles Griot).

For whole-cell recordings, glass pipettes were filled with an internal solution containing (in mM): 120 K-gluconate, 15 KCl, 0.5 Na₂EGTA, 2 Mg₂ATP, 0.4 Na₂GTP, 10 HEPES, and 5 Na₂-phosphocreatine (adjusted to pH 7.2 with KOH; adjusted to 290 mOsm with sucrose). Neurons were voltage clamped at −60mV to record EPSCs in the presence of picrotoxin (50 μM). All current clamp intrinsic excitability experiments were performed with CNQX (cyanquixaline (6-cyano-7-nitroquinoxaline-2,3-dione) (10 μM) and picrotoxin (50 μM). Each cell was held around −65 mV and we used a current step protocol to determine spike output and FSL. The step protocol consisted of a 0.5 s–40 pA hyperpolarizing pulse, followed by increasing 1 s square steps from 0 to +30 pA in 5 pA increments. Spikes were detected using a threshold search in Clampfit. FSL was calculated from the point of the depolarizing step initiation to the peak of the first spike.

Extracellular electrical stimulation was delivered using a monopolar glass electrode filled with aCSF. Biphasic paired pulse stimulations were delivered at 30–100 μA at 20 ms intervals. Trains of electrical stimulations to evoke GCaMP6s responses were delivered at 100 μA at 10 Hz for 5 s.

Electrophysiological recordings were collected with a Multiclamp 700B amplifier (Molecular Devices), filtered at 2 kHz, and digitized using the Digidata 1440a (Molecular Devices). sEPSC currents were analyzed using Mini Analysis

and all other electrophysiological data were analyzed with Clampfit 10 (Molecular Devices). GCaMP6s images were acquired using Fluoview 1000 at ~2 Hz frame rate and analyzed using Image J. Changes in fluorescence ($\Delta F/F$) were calculated, where $F$ is the averaged baseline fluorescence for each region of interest.

**Blood collection and ELISA**. Tail blood samples were collected via heparinized capillary tubes. All mice were previously habituated to handling for at least 7 consecutive days. Plasma corticosterone was measured using an ELISA (Arbor Assays) according to the manufacturer's instructions. For ACTH measurements, mice were decapitated and trunk bloods were collected in lavender EGTA-coated tubes. All samples were kept on ice and centrifuged at 4 °C within minutes of collection. ACTH was measured using an ELISA (MD Bioproducts) according to the manufacturer's instructions.

**Immunohistochemistry**. GCaMP6s was labeled in fixed coronal brain sections (30 μm) with a GFP antibody (chicken anti-GFP; 1:3000; Aves Labs) and visualized using Alexa Fluor 488 goat anti-chicken IgG (1:500; Molecular Probes, Life Technologies). Sections were imaged under confocal microscopy (Zeiss LSM 710) and analyzed using Image J to quantify GCaMP and CRH-tdTomato colocalization. We observed GCaMP6s transfection in 58.1 ± 2.1% of CRH neurons in the ipsilateral PVN and 88.0 ± 1.8% of GCaMP6s-transfected cells were positive for CRH-tdTomato (Fig. 1a and Supplementary Table 1).

**Data analysis and statistics**. Photometry data were processed using Prism and Excel to calculate linear regression and $\Delta F/F$ changes. We did not correct for any movement artifacts as they had minimal impact on the overall recording. Brain slice calcium images were acquired and analyzed using Fluoview 1000 and Image J, where regions of interest were drawn around each CRH neuron. For in vitro GCaMP6s imaging, corrections for photobleaching were not necessary and $\Delta F/F$ changes were normalized to the baseline.

For calculating correlations in spike to $\Delta F/F$ change in vitro, total spike counts were collected from each phasic burst-firing window using the pClamp 10 threshold search. Each phasic burst window was defined as a period where GCaMP6s fluorescence was elevated (>10% $\Delta F/F$) and returned to baseline levels. Total accumulation of GCaMP6s fluorescence during each burst was correlated with the spike count using Prism. Spontaneous synaptic currents and photometry GCaMP6s transients were detected using MiniAnalysis.

All data are presented as mean ± SEM in the figures and text. All group comparisons for photometry data were performed using RM multiple comparisons two-way ANOVA (Holm–Sidak post hoc test), unless otherwise stated. All statistical analyzes were performed using Prism. *$P < 0.05$, **$p < 0.01$, and ***$p < 0.001$.

**Reporting summary**. Further information on research design is available in the Nature Research Reporting Summary linked to this article.

## Data availability
All datasets supporting the findings of this study are available upon reasonable request.

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

## Acknowledgements

We are extremely grateful to Professor Allan Herbison for assistance establishing the fiber photometry technique and for valuable input throughout the project. We also thank Tussock Innovation and Leo van Rens (EMTech, University of Otago) for technical assistance. We thank Dr. Michel Herde, Ms. Caroline Focke, and Dr. Emmet Power for comments on an earlier version of this manuscript. This work was funded by the Prime Minister's MacDiarmid Emerging Scientist Prize and the Neurological Foundation Small Project Grant awarded to K.J.I. J.S.K. was supported by a New Zealand Lottery Health Research Postdoctoral Fellowship. K.J.I. was supported by a Sir Charles Hercus Health Research Council Fellowship.

## Author contributions

J.S.K and K.J.I designed research. J.S.K performed research. S.Y.H contributed analytic tools. J.S.K and K.J.I analyzed data. All authors contributed to writing the manuscript.

## Competing interests

The authors declare no competing interests.
