## [Transparent Peer Review File · Nature Communications]

Reviewers' comments:

Reviewer #1 (Remarks to the Author):

This study examined the population activity of CRH neurons in the paraventricular nucleus of the hypothalamus (PVN) in freely moving male mice using fiber photometry. Authors investigated the adaptations of stress-evoked activities of CRH neurons and the roles of corticosterone (CORT) in the CRH neurons activity. Both endogenous and exogenous CORT had little suppressive effects on the rapid activation during the stress exposure. However, CORT suppressed the baseline and low-level activities that persisted after the termination of stress. The rapid CRH activity habituated to the same but not to a novel stressor, consistent with the hormonal habituation reported by earlier studies. The relative lack of CORT contributions to the stress-evoked, rapid CRH neurons activity is new and very interesting, as the effects of CORT on CRH neurons activity in vivo remains unknown. The data is technically sound. However, there are a few issues that require clarification to strengthen the main conclusion of the paper. In particular, the data revealed characteristic temporal dynamics of CRH neuron population activity that were quantified as different readouts (peak, mean, cumulative integrated, and baseline $\Delta F/F$), and these readouts have different sensitivity to CORT and stress experience. Based on the data, the authors proposal that "stress experience and stress hormones modulate distinct components of CRH neuronal activity (ex. from the last sentence in the abstract)". However, the downstream physiological consequences of such "distinct components of CRH neuronal activity" remains obscure in the paper.

Major comments:

- 1) The relationship between CRH neurons' Ca^{2+} elevation and ACTH levels is not as straightforward as authors propose. In page 10, line 240, authors describe that "These results suggest that fast CORT negative feedback powerfully suppresses pituitary secretion without dramatically impacting the magnitude of CRH neuronal responses to threat." However, the data in Fig. 5 do not fully support this conclusion. While CORT injection did not suppress the mean $\Delta F/F$ (Fig. 5D), it significantly suppressed the cumulative integrated $\Delta F/F$ (Fig. 5E). This appears to be due to the difference in the sustained $\Delta F/F$ elevation (Fig. 5A). Why does the mean $\Delta F/F$ overrule the cumulative $\Delta F/F$? There is no experimental data or references to support the relative importance of the mean over the cumulative $\Delta F/F$. Indeed, the magnitude of the CORT-induced ACTH suppression is comparable to that of the cumulative $\Delta F/F$ suppression. This is important for a major proposal of this paper regarding the "disconnect" between CRH neurons' activity and the HPA axis outputs. (page 15, line 353-362).
- 2) Along the same line, the relationship between CRH $\Delta F/F$ elevation and ACTH release would deserve to be examined in the repeated WIN exposure experiments (ex. Fig. 2, Fig. 3).
- 3) Fig. 5F. The baseline ACTH should be reported for both vehicle and CORT groups. This is important as all mice are pretreated with MET.
- 4) Page 7, line 134. "In response to WIN2, metyrapone treatment did not alter the adaptive responses to WIN2." This statement is not supported by the data shown in Fig. 2 and the description following the sentence. There was a significant increase by metyrapone in the cumulative $\Delta F/F$ after WIN2 (Fig. 2H). Also Fig. 2C show a similar effects of MET on the peak $\Delta F/F$. This was also the case in Fig. 3H. What are the differences between the peak, mean and cumulative $\Delta F/F$ elevation with regard to CRH neurons' activity and their downstream consequences? Later in the text (Page 12, line 302-205), the peak response was interpreted as startle response which was dissociable from the habituation of the 5-min average response by repeated WIN exposures.
- 5) The effects of CORT on the tonic $\Delta F/F$. In Fig. 3, authors describe that MET caused a sustained Ca^{2+} elevation, preventing the return of the $\Delta F/F$ to the baseline that was observed in the vehicle control (page 7, line 155 for example, "However, in metyrapone-treated mice lacking the ability to

synthesize CORT de novo, tonic CRH activity remained elevated"). Based on this interpretation, it is expected that in the experiments shown in Fig. 4, 5 where all mice were pre-injected with MET, vehicle group (MET alone) shows a sustained, post-stress Ca²⁺ elevation when compared to CORT group. However, this was not the case. The exogenous CORT decreased the tonic $\Delta F/F$ below non-CORT group whereas the CORT injection had no effects on the post-stress sustained Ca²⁺ elevation (Fig. H, I). In this regard, the interpretation of the data shown in Fig. 3 and Fig. 4.5 are somewhat inconsistent.

6) Related to above, the cumulative $\Delta F/F$ (Fig. 3D) and heat map (Fig. 3B) shows that the $\Delta F/F$ became below baseline (a decline in the cumulative curve in Fig. 3D and negative values in Fig. 3B). To make Fig. 3 and Fig. 4-5 data comparable, the baseline shift after the first stress may need to be corrected prior to the 2nd stress.

7) While Supplementary Figure 1 support the idea that the Ca²⁺ signals reflect CRH neurons' firing activity, this correlation was examined in slices treated with noradrenaline. Do firing activity and Ca²⁺ elevations correlate in the absence of noradrenaline? Noradrenaline has complex pre and postsynaptic actions in the PVN (Neuroscience. 2000;96(4):743-51). In other neurons, it has been reported to have direct postsynaptic actions (J Neurosci. 1998 Dec 15;18(24):10619-28) and induce Ca²⁺ changes (J Physiol. 2005 Jan 15; 562(Pt 2): 553–568). Thus, this particular experimental condition may overestimate the coupling between action potential firing and Ca²⁺ elevation. This issue needs to be clarified.

8) Fig. 2C and Fig. 3H. Did the two-way ANOVA reveal significant interaction? While the 5 min bin averages do not show MET effects on WIN2 response (Fig.2E, Fig. 3E) MET appears to prevent the diminution of the peak $\Delta F/F$.

9) More generally, all two-way ANOVA should report p values for interaction, main effects in addition to post-hoc comparison.

10) Authors show that, in slice experiments, bath application of CORT had no effects on eEPSCs but replicated its actions on sEPSCs. Based on these results, it is proposed that "This may provide one potential mechanism to explain the in vivo observations where tonic CRH activity is suppressed by negative feedback but stress-activity is not." Page 11, line 265-266. In this experiment, do you observe a rapid decrease in the baseline/tonic GCaMP6s fluorescence elevations?

11) Is the habituation of GCaMP6s elevations to daily WIN stress independent of CORT? The 30 min and 120 min interval experiments may hint it is the case, but the 24 h interval would allow for much slower actions of CORT leading to potentially long-lasting neural plasticity. This was partly touched in the discussion (Page 15, line 363-371) but mainly focuses on changes in CRH transcriptional activity.

Minor:

1) Fig. 6. The cumulative $\Delta F/F$ should also be reported. Fig 6E shows that the sustained $\Delta F/F$ elevation after the foot shock was suppressed by the prior exposure to WIN.

2) Page 11, line 261. Unfinished sentence should be removed. "As noradrenaline is considered to"

3) Page 27, line 702. In Fig. 5C it is indicated " $*p<0.05$ " but the graph only has *** which I assume indicate a smaller p value.

4) The reference channel (405 nm, actual plot and the one after the linear regression correction) should be shown in Supplementary data.

Reviewer #2 (Remarks to the Author):

The manuscript entitled "Stress experience and hormone feedback tune distinct components of CRH neuron activity" employs an elegant approach to "fact check" some long held assumptions and challenges the current dogma regarding negative steroid hormone feedback on the HPA axis. Further, data presented in this manuscript elegantly demonstrates and confirms CRH neuron habituation as a long-term adaptation of the neuroendocrine stress response. These data are compelling and are transformative to the field. However, many of the findings are observational/descriptive, leaving many mechanistic questions unanswered.

1. These data suggest that CRH neurons process stimuli (different stressors) differently. However, the mechanism whereby CRH neurons distinguish between types of stressors remains unclear.
2. The authors demonstrate variability in CRH neuron activity after 5 min of white noise stress (Figure 1). Did this variability correlate with CORT levels or the time course of the CORT response in individual animals?
3. The authors should discuss the translational impact of dissociating the baseline regulation of CRH neurons vs. the evoked/stress-induced regulation. Does this suggest that the stress-induced regulation of CRH neurons and the diurnal regulation can be dissociated? Does this suggest that we should revisit targeting CRH neurons/HPA axis for therapeutics?
4. The baseline activity of CRH neurons diverges from baseline activity ~40 mins following CORT administration and the peak wasn't observed until 75 min. How does this relate to CORT exposure levels (PK)?
5. The authors suggest that their data demonstrate that CRH neurons are capable of scaling their output based on stress modality/severity. How?
6. The authors should discuss the importance of how their data support synaptic regulation of CRH neurons and the HPA axis rather than relying largely on neuroendocrine feedback mechanisms.
7. A diagram of tonic vs. stress-induced regulation of CRH neurons and the HPA axis would be beneficial.

Reviewer #3 (Remarks to the Author):

In this manuscript, Kim et. al. carefully describe patterns of hypothalamic CRH neuron calcium signals in vivo after novel and repeated stressors, as well as after pharmacological manipulation of CORT. The data are clear and, for the most part, the reported differences appear robust. A number of points for consideration are listed below. My major concern is that, as elegant as the studies are, they are highly descriptive in nature. Outside of the somewhat unanticipated CORT results, much of the data, though beautiful, re-demonstrate the association of hypothalamic CRH neuron activity with stress, and that this activity is habituated via repeating the same stressor. There seem to be many points at which the authors could dig in and perform interventional experiments to test their mechanistic ideas, and I have indicated some of these below. Given the recent Kim et al. Nature Neuroscience paper (admittedly very recent) on a very similar topic in which interventional experiments were performed, this seems all the more important.

Points for consideration:

- 1) In the first paragraph of the introduction, the authors state: "However, the plasticity mechanisms driving adaptations in these neural circuits following stress remain poorly studied". A similar statement is made in line 44-45. I have two significant concerns here. First, to my mind this is not true, as the labs of Bains and Tasker, for example, have extensively studied plasticity in the PVN as it relates to stress and their work should be more explicitly mentioned in this context within the intro and discussion. Second, this statement seems to set up the notion that the authors

will be shedding light on this issue in the current manuscript, yet I see no experiments that address plasticity per se in this system.

- 2) The CORT resupply experiments are interesting, but discussion of the dosage chosen needs to be provided to give context on the physiologic versus supra-physiological relevance here.
- 3) In Figure 3 the authors hypothesize a mechanism involving “presumably loss of endogenous negative feedback” (line 151). This is the sort of statement that could be directly tested in a more mechanistically-oriented manuscript.
- 4) Similarly lines 265-66 speculate on potential mechanisms that could be tested.
- 5) In line 373, the authors propose CORT strongly inhibits the CRH stress response, as opposed to stating it desensitized or some other mechanism. Again, this could be directly tested.
- 6) In the description of the data in Figure 2H, there’s discussion of divergence of the curves at 3.5 minutes, but this is not indicated on the graph. Moreover, two bars with different asterisks are printed over the plots, but it’s not clear from the legend or results what they refer to. This presentation strategy is used throughout.
- 7) The visual differences in the mean signals in Figure 3A seem heavily driven by just a few sweeps in the MET heatmap. The differences in these heatmaps seem somewhat subtle, and might be improved with a short timescale of presentation, since the analysis in 3C focuses on a short timescale.
- 8) In Figure 3J, the far-right bar has two points that are very low relative to the others. Were the deposits and fiber placements verified in cases such as these?
- 9) The differences in Figure 4A through D are extremely subtle and one worries about the influences of baseline drift, etc. Do the authors observe washout of this CORT effect?

Minor issues:

- 1) Throughout, the authors simply refer to “CRH neurons”. This needs to be specified as hypothalamic CRH neurons given the heavy expression of this neuropeptide throughout extrahypothalamic cortical and subcortical structures responsive to stress.
- 2) Line 108: “novel” would be more appropriate than “external” here.
- 3) Lines 216-17: Provide citations for these time windows.
- 4) Line 261: sentence fragment on norepinephrine should be removed.
- 5) Line 374: “adaptive plasticity” is stated, but plasticity itself was not directly explored in this manuscript.

Reviewer #4 (Remarks to the Author):

In this work, the authors use in vivo fiber photometry to record Ca²⁺ signals from GCaMP6s-expressing CRH neurons in the hypothalamus of animals exposed to a stressor (in this study very loud white noise or footshock). These Ca²⁺ signals can be used as a surrogate for neuronal action potential firing, and represent a mean population signal originating from all the somata, dendrites and axons located in the volume excited by the blue wavelength light. These experiments are very challenging, mainly because they were performed in freely-moving mice and because the hypothalamus is a very deep and hard-to-access brain structure. The in vivo fiber photometry experiments using two sinusoidally-modulated excitation wavelengths (405 nm being the wavelength where GCaMP fluorescence is Ca²⁺ insensitive and 465 nm the wavelength where GCaMP fluorescence is dependent on the cytosolic Ca⁺ concentration) are well designed and state-of-the-art. I have a few questions/comment that should be, in my opinion, addressed in a revised manuscript:

- 1) The authors should clarify some basic properties of the fiber photometry system used here addressing the following questions: (1) What is the numerical aperture of the fiber used? (2) also, based on this feature, what is the (estimated) excitation volume of your system? (3) how many cells, including their axons/dendrites, are located in this volume?
- 2) In fig. S2, the authors show mean responses recorded from GCaMP6s- and GFP-expressing

animals. The GFP expressing animals show almost no fluctuations. This result is evidence for the lack of any motion artifacts during the experiments, even when the animals are exposed to the stressor. However, these signals were collected from a different group of animals, and the number of data points is low ($n=3$). Instead, showing the 405 nm signal, the isosbestic wavelength of GCaMP6s, that was in any case recorded in the GCaMP6s-expressing animals would make the argument stronger. It would also be very informative since the 405 nm signal is used for the dF/F calculation. By the way, the authors mention the 405 nm reference channel in line 85 on page 5, but never show the actual data. Also, please clarify whether there was any data excluded due to motion artifacts.

3) As shown, for example, in fig. 1j, the response to the white noise is biphasic and consists of an initial fast and higher-amplitude response and a sustained, but lower-amplitude response. Are these different "type" of responses caused by the same or different cell populations? Is there, for example, subsequent activation of new neurons by the initially responding neurons? It is interesting in this context that during repeated exposures over 4 days (figure 7) the peak response remains unchanged, whereas the sustained response is reduced.

4) Along the same lines: fig. 3i2 shows that after metyrapone application, there is an increase in the amplitude of the Ca^{2+} transients, but not in their frequency. It would be interesting to either experimentally address or at the very least discuss the possible mechanisms underlying this finding. Is there evidence for recruitment of neurons into the active neuron pool or do the same neurons "simply" fire more action potentials?

Minor points:

1) Please clarify in the figure legends what animals are included in fig. 2g.

2) It is not clear to me how the cumulative integrated df/f , for example in fig. 2h, was computed. I would assume from the data that in the first 4 seconds after WN2 onset, there should be an increase in the cumulative integrated df/f to 0.89.

3) page 8, line 180: I think this should be fig. 3H.

Response to Reviewers' comments:

Reviewer #1:

This study examined the population activity of CRH neurons in the paraventricular nucleus of the hypothalamus (PVN) in freely moving male mice using fiber photometry. Authors investigated the adaptations of stress-evoked activities of CRH neurons and the roles of corticosterone (CORT) in the CRH neurons activity. Both endogenous and exogenous CORT had little suppressive effects on the rapid activation during the stress exposure. However, CORT suppressed the baseline and low-level activities that persisted after the termination of stress. The rapid CRH activity habituated to the same but not to a novel stressor, consistent with the hormonal habituation reported by earlier studies. The relative lack of CORT contributions to the stress-evoked, rapid CRH neurons activity is new and very interesting, as the effects of CORT on CRH neurons activity in vivo remains unknown. The data is technically sound. However, there are a few issues that require clarification to strengthen the main conclusion of the paper. In particular, the data revealed characteristic temporal dynamics of CRH neuron population activity that were quantified as different readouts (peak, mean, cumulative integrated, and baseline $\Delta F/F$), and these readouts have different sensitivity to CORT and stress experience. Based on the data, the authors proposal that “stress experience and stress hormones modulate distinct components of CRH neuronal activity (ex. from the last sentence in the abstract)”. However, the downstream physiological consequences of such “distinct components of CRH neuronal activity” remains obscure in the paper.

We thank the reviewer for their thorough assessment of our manuscript. As noted below, we have provided additional clarification regarding our different measurements of CRH activity. We have also added additional discussion (lines 386-428) on the potential physiological importance of the different components of CRH neural activity following stress.

Major comments:

1) The relationship between CRH neurons' Ca^{2+} elevation and ACTH levels is not as straightforward as authors propose. In page 10, line 240, authors describe that “These results suggest that fast CORT negative feedback powerfully suppresses pituitary secretion without dramatically impacting the magnitude of CRH neuronal responses to threat.” However, the data in Fig. 5 do not fully support this conclusion. While CORT injection did not suppress the mean $\Delta F/F$ (Fig. 5D), it significantly suppressed the cumulative integrated $\Delta F/F$ (Fig. 5E). This appears to be due to the difference in the sustained $\Delta F/F$ elevation (Fig. 5A). Why does the mean $\Delta F/F$ overrule the cumulative $\Delta F/F$? There is no experimental data or references to support the relative importance of the mean over the cumulative $\Delta F/F$. Indeed, the magnitude of the CORT-induced ACTH suppression is comparable to that of the cumulative $\Delta F/F$ suppression. This is important for a major proposal of this paper regarding the “disconnect” between CRH neurons' activity and the HPA axis outputs. (page 15, line 353-362).

We apologise for the lack of clarity between cumulative and mean $\Delta F/F$ changes. We have used cumulative $\Delta F/F$ analysis to detect changes that manifest slowly over longer timeframes. We have clarified this in our text (lines 131-133).

We have also clarified some of our wording when referring to effects on CRH neuron stress responses. Specifically, effects on stress evoked activity refer to effects that manifest during the 5 minutes of stress, whereas effects on post-stress activity refer to effects that manifest during the 5 minute period post stress. Overall activity refers to both during and post-stress. Tonic activity refers to effects that manifest >5 minutes post stress.

With these points in mind, we interpret the data in Figure 5 as showing that pre-treatment with CORT 30 min prior to WN has no effect on the stress evoked activity, however, does

result in a significant reduction in the cumulative $\Delta F/F$ that only manifests in the post-stress period.

Lastly, we provide for the reviewer photometry data obtained from mice pre-treated with metyrapone, without white noise (which is the same treatment given to the mice for which baseline ACTH samples were obtained). We calculated the cumulative $\Delta F/F$ and plotted it against the existing data, which serves as the photometry correlate of our “basal” ACTH group. As can be seen, even at baseline levels of CRH neuron activity (when the mean cumulative $\Delta F/F$ is close to zero) there is a basal secretion of ACTH (40.6 pg/mL). If we subtract the basal level of ACTH away from the stress evoked peak, then we obtain the change in ACTH induced by stress (Veh control Δ ACTH = 138.2 pg/mL; CORT Δ ACTH = 51.5 pg/mL). When this is taken into consideration, the suppression of stress evoked ACTH is 63%. However, the cumulative $\Delta F/F$ is only reduced by 28% (and the main group effects ANOVA shows no significant effect). We hope this clearly illustrates that the magnitude of the ACTH suppression is much larger than the suppression of cumulative $\Delta F/F$ at the same time point.

2) Along the same line, the relationship between CRH $\Delta F/F$ elevation and ACTH release would deserve to be examined in the repeated WIN exposure experiments (ex. Fig. 2, Fig. 3).

This is an excellent idea, however, we cannot perform repeated ACTH measurements in mice as it requires decapitation to retrieve the amount of plasma required for the ELISA.

3) Fig. 5F. The baseline ACTH should be reported for both vehicle and CORT groups. This is important as all mice are pretreated with MET.

We agree with the reviewer that baseline ACTH levels are important, particularly in MET treated animals. However as explained above, this was not possible due to the need to obtain large volume blood samples. There is a “Basal” sample in Fig 5F which comes from MET-pretreated mice without white noise exposure. We have changed the group identify to “no WN” and clarified this in the text (lines 238-240).

4) Page 7, line 134. “In response to WIN2, metyrapone treatment did not alter the adaptive responses to WIN2.” This statement is not supported by the data shown in Fig. 2 and the description following the sentence. There was a significant increase by metyrapone in the cumulative $\Delta F/F$ after WIN2 (Fig. 2H). Also Fig. 2C show a similar effects of MET on the peak $\Delta F/F$. This was also the case in Fig. 3H. What are the differences between the peak, mean and cumulative $\Delta F/F$ elevation with regard to CRH neurons’ activity and their downstream consequences? Later in the text (Page 12, line 302-205), the peak response was interpreted as startle response which was dissociable from the habituation of the 5-min average response by repeated WIN exposures.

As noted in response to the reviewers first comment, we have now more clearly defined CRH neuron responses into stress-evoked and post-stress periods. We have now modified the statement that was originally on page 7, line 134 to read “Mean CRH activity during WN2 was also not different between vehicle and metyrapone treatment groups.” Importantly, the significant difference between Veh and MET groups in our cumulative $\Delta F/F$ analysis is only revealed 3.5 min after WN2 ends.

The reviewer raises an interesting point regarding the peak $\Delta F/F$. We have seen this discrepancy between Veh and MET groups on both occasions. As noted in the manuscript, we think that the peak $\Delta F/F$ at WN onset is driven by a startle response. Our strongest evidence for this comes from the 4 day WN experiment (Fig 8). Here we observe a marked habituation of the mean $\Delta F/F$ response to WN but no significant habituation of the peak response at WN onset. We believe that this data shows that these two components of stress evoked CRH neural activity can be regulated independently.

Physiologically, we speculate that the startle response would be important for quickly activating CRH neurons when exposed to an unexpected potential threat. Subsequently, as the animal has time to determine the nature of the threat, the stress response can then be adjusted to the level of danger (discussion lines 408-413). Consistent with this idea, recent work published in Nature Neuroscience (Kim et al, 2019) demonstrate that CRH neurons have a fast initial onset followed by a slower second phase of sustained activity (which varies dependent on stress modality).

5) The effects of CORT on the tonic $\Delta F/F$. In Fig. 3, authors describe that MET caused a sustained Ca^{2+} elevation, preventing the return of the $\Delta F/F$ to the baseline that was observed in the vehicle control (page 7, line 155 for example, “However, in metyrapone-treated mice lacking the ability to synthesize CORT de novo, tonic CRH activity remained elevated”). Based on this interpretation, it is expected that in the experiments shown in Fig. 4, 5 where all mice were pre-injected with MET, vehicle group (MET alone) shows a sustained, post-stress Ca^{2+} elevation when compared to CORT group. However, this was not the case. The exogenous CORT decreased the tonic $\Delta F/F$ below non-CORT group whereas the CORT injection had no effects on the post-stress sustained Ca^{2+} elevation (Fig. H, I). In this regard, the interpretation of the data shown in Fig. 3 and Fig. 4.5 are somewhat inconsistent.

We have clarified two points in the text.

On lines 150-155 we have reworded our description of the effects on tonic activity. We now note that almost all animals exposed to negative feedback show tonic activity levels that are either the same or below baseline. Whereas animals without negative feedback show tonic activity that are either the same or above baseline. We have also modified the text making it clearer that we are referring to the cumulative $\Delta F/F$ changes.

However we disagree that our interpretation of cumulative $\Delta F/F$ during tonic activity is inconsistent between Figures 3-5. It is important to note that tonic activity starts to diverge 30 minutes post-stress in Figure 3. Experiments shown in Figure 5 are simply not long enough to show this clear divergence. The divergence can however be clearly seen in Figure 4 when we record for 2 hours post stress.

Lastly, to make comparisons between Figure 3D and 4D easier, in the revised manuscript we now include the injection stress response in the cumulative $\Delta F/F$ trace (previously the response to injection was excluded from the cumulative integral analysis in the original Fig 4D). It can now be more clearly seen that in the vehicle group, following injection, activity remains elevated (and even slightly increases) over the 2 hour period.

Regarding the reviewer’s comments on Fig 4 H and I: we are showing in these panels that CORT injection does not affect the stress evoked activity as we believe this is a separate

component from the tonic activity. In these panels, the data have been corrected for the sustained shift in baseline activity. This is why there is no difference in the baseline following stress in panel H and I.

6) Related to above, the cumulative $\Delta F/F$ (Fig. 3D) and heat map (Fig. 3B) shows that the $\Delta F/F$ became below baseline (a decline in the cumulative curve in Fig. 3D and negative values in Fig. 3B). To make Fig. 3 and Fig. 4-5 data comparable, the baseline shift after the first stress may need to be corrected prior to the 2nd stress.

We provide for the reviewer the baseline normalised data. We have not included this in the revised manuscript as it does not alter the statistics or our conclusions from this experiment.

7) While Supplementary Figure 1 support the idea that the Ca^{2+} signals reflect CRH neurons' firing activity, this correlation was examined in slices treated with noradrenaline. Do firing activity and Ca^{2+} elevations correlate in the absence of noradrenaline? Noradrenaline has complex pre and postsynaptic actions in the PVN (Neuroscience. 2000;96(4):743-51). In other neurons, it has been reported to have direct postsynaptic actions (J Neurosci. 1998 Dec 15;18(24):10619-28) and induce Ca^{2+} changes (J Physiol. 2005 Jan 15; 562(Pt 2): 553–568). Thus, this particular experimental condition may overestimate the coupling between action potential firing and Ca^{2+} elevation. This issue needs to be clarified.

We thank the reviewer for identifying this potential confound. We used noradrenaline to induce spiking as CRH neurons are typically silent in acute brain slices. We have repeated this experiment with a low concentration (7.5mM) KCl in the aCSF and attached the data for review purposes. We observed equally reliable correlations between GCaMP6s fluorescence and spiking activity. Our modified Supplementary Figure now includes the combined data from both the noradrenaline and KCl experiment.

8) Fig. 2C and Fig. 3H. Did the two-way ANOVA reveal significant interaction? While the 5 min bin averages do not show MET effects on WIN2 response (Fig.2E, Fig. 3E) MET appears to prevent the diminution of the peak ΔF/F.

We have reported the *p*-values for interaction. We agree with the reviewer that MET treatment attenuated the reduction of the peak ΔF/F and our post-hoc *p*-values reflect this also. However the ANOVA did not reveal a significant interaction as the overall outcome is that WN2 peak (regardless of treatment group) is still lower than their respective WN1 peaks.

9) More generally, all two-way ANOVA should report *p* values for interaction, main effects in addition to post-hoc comparison.

We have now included *p*-values for interaction and main effects for both groups and timepoints for all two-way ANOVA analyses in the figure legends.

10) Authors show that, in slice experiments, bath application of CORT had no effects on eEPSCs but replicated its actions on sEPSCs. Based on these results, it is proposed that “This may provide one potential mechanism to explain the in vivo observations where tonic CRH activity is suppressed by negative feedback but stress-activity is not.” Page 11, line 265-266. In this experiment, do you observe a rapid decrease in the baseline/tonic GCaMP6s fluorescence elevations?

Unfortunately we see almost no spontaneous spiking activity in acute brain slices so have been unable to test the effect of CORT on baseline/tonic GCaMP6s fluorescence elevations.

11) Is the habituation of GCaMP6s elevations to daily WIN stress independent of CORT? The 30 min and 120 min interval experiments may hint it is the case, but the 24 h interval would allow for much slower actions of CORT leading to potentially long-lasting neural plasticity. This was partly touched in the *discussion* (Page 15, line 363-371) but mainly focuses on changes in CRH transcriptional activity.

The reviewer raises an interesting experimental idea. However, due to the limited time window to submit revisions, we were unable to perform this experiment.

Minor:

1) Fig. 6. The cumulative ΔF/F should also be reported. Fig 6E shows that the

sustained $\Delta F/F$ elevation after the foot shock was suppressed by the prior exposure to WIN.

This has now been added.

2) Page 11, line 261. Unfinished sentence should be removed. "As noradrenaline is considered to"

This has been corrected.

3) Page 27, line 702. In Fig. 5C it is indicated " $p < 0.05$ " but the graph only has *** which I assume indicate a smaller p value.

This has been corrected.

4) The reference channel (405 nm, actual plot and the one after the linear regression correction) should be shown in Supplementary data.

We now show the 405 nm traces side by side in our revised Supplementary Figure 2. This is shown as % change as we wanted to show the two signals in the same scale.

We have included below the mean and raw data of all the 405 traces (n=64) for review purposes only. Bleaching over a 30min period (white noise onset at 10min) is minimal. ΔRFU = change in raw fluorescence units.

Reviewer #2:

The manuscript entitled “Stress experience and hormone feedback tune distinct components of CRH neuron activity” employs an elegant approach to “fact check” some long held assumptions and challenges the current dogma regarding negative steroid hormone feedback on the HPA axis. Further, data presented in this manuscript elegantly demonstrates and confirms CRH neuron habituation as a long-term adaptation of the neuroendocrine stress response. These data are compelling and are transformative to the field. However, many of the findings are observational/descriptive, leaving many mechanistic questions unanswered.

We thank the reviewer for their comments and agree that our original manuscript lacked some mechanistic insights. We have now taken an electrophysiological approach to further understand CORT-induced plasticity that underlies the suppressions in tonic activity. These data are described in our revised manuscript and can be found in the new Figure 6.

1. These data suggest that CRH neurons process stimuli (different stressors) differently. However, the mechanism whereby CRH neurons distinguish between types of stressors remains unclear.

It is an interesting observation that a shorter stress in the form of footshock elicits stronger CRH neuron activity than a 5 minute white noise. This processing of stimuli is unlikely to be driven by CRH neurons themselves, but rather their inputs. While we have future plans to study such mechanisms, we were unable to do so for this revision.

2. The authors demonstrate variability in CRH neuron activity after 5 min of white noise stress (Figure 1). Did this variability correlate with CORT levels or the time course of the CORT response in individual animals?

We have included for the reviewer the data analysed from our 4-day habituation experiment (from group data in Fig 8). This new analysis shows that mean post-stress activity (5 min time window following stress) shows a similar correlation with CORT levels as mean stress activity shown in Fig 8F.

In response to the reviewers second question regarding response variability and the time course of CORT responses- unfortunately we are unable to take repeated CORT samples at the frequency required to study the time course of CORT responses. This is mainly because handling stress remains an issue.

3. The authors should discuss the translational impact of dissociating the baseline regulation of CRH neurons vs. the evoked/stress-induced regulation. Does this suggest that the stress-induced regulation of CRH neurons and the diurnal regulation can be dissociated? Does this suggest that we should revisit targeting CRH neurons/HPA axis for therapeutics?

Given the current therapeutic options, we don't think it would be possible to differentially modulate baseline CRH activity vs stress induced activation without potentially impacting other systems (eg learning and memory, appetite etc). Therefore, we are hesitant to suggest that these findings could have current therapeutic implications.

Regulation of baseline activity as a mechanisms of diurnal control of the CRH neuron activity is an interesting idea. However, the limited evidence that is available suggests that CRH neural activity is increasing towards the end of the inactive (light) period coincident with an increasing level of CORT negative feedback (which would suppress CRH activity). The net effect on CRH activity patterns would be complex and hard to predict. For this reason, we have not specifically discussed diurnal regulation in the revised manuscript.

4. The baseline activity of CRH neurons diverges from baseline activity ~40 mins following CORT administration and the peak wasn't observed until 75 min. How does this relate to CORT exposure levels (PK)?

Data presented in Supplementary Figure 3 shows that following CORT injection, plasma levels peak within 15 minutes and then steadily decline over the subsequent 45 minutes. Previous work from Droste et al (2008 Endocrinology) has shown that hippocampal CORT levels are also elevated 15 min after a peripheral CORT injection, however, do not peak until 35 min following the injection. Importantly, brain CORT was shown to remain elevated for over an hour following a single peripheral injection. We anticipate a similar profile for hypothalamic CORT in our mice.

We observe fast (presumably non-genomic) effects of CORT on sEPSC frequency in brain slices by 8 minutes. We expect this suppression to persist as long as brain CORT remains elevated. However, if this sEPSC effect was solely responsible for the suppression of tonic activity, we would have expected a suppression of tonic activity to manifest much more quickly than what was observed during our photometry recordings. Our subsequent experiments have revealed that changes in intrinsic excitability manifest with a time course that matches CORT suppression of tonic activity. The time course of these changes, and the fact that they persist following cutting of brain slices, suggest that they are due to genomic CORT signalling.

Therefore, while we believe that fast CORT mediated suppression of sEPSC frequency is likely to occur in vivo, it appears that slower CORT mediated suppression of intrinsic excitability matches more closely with the in vivo suppression of tonic activity that appears 40 min after CORT injection. These points are now noted in the results (lines 300-303) and discussion (lines 386-399).

5. The authors suggest that their data demonstrate that CRH neurons are capable of scaling their output based on stress modality/severity. How?

We show in Figure 7 that brief footshock stress can induce longer lasting CRH neuron activity than a 5 min white noise stress. However, we appreciate that our statement regarding scaling of output is only based on comparison of two stressors. For this reason we have decided to remove the statement from the current revised manuscript.

6. The authors should discuss the importance of how their data support synaptic regulation of CRH neurons and the HPA axis rather than relying largely on neuroendocrine feedback mechanisms.

We have added this idea to our discussion and this can be found in lines 415-420.

7. A diagram of tonic vs. stress-induced regulation of CRH neurons and the HPA axis would be beneficial.

We have added a summary diagram to the supplementary material (Supplementary Figure 5).

Reviewer #3:

In this manuscript, Kim et. al. carefully describe patterns of hypothalamic CRH neuron calcium signals in vivo after novel and repeated stressors, as well as after pharmacological manipulation of CORT. The data are clear and, for the most part, the reported differences appear robust. A number of points for consideration are listed below. My major concern is that, as elegant as the studies are, they are highly descriptive in nature. Outside of the somewhat unanticipated CORT results, much of the data, though beautiful, re-demonstrate the association of hypothalamic CRH neuron activity with stress, and that this activity is habituated via repeating the same stressor. There seem to be many points at which the authors could dig in and perform interventional experiments to test their mechanistic ideas, and I have indicated some of these below. Given the recent Kim et al. Nature Neuroscience paper (admittedly very recent) on a very similar topic in which interventional experiments were performed, this seems all the more important.

We thank the reviewer for their thoughtful comments. To address the reviewers concern regarding mechanistic insight, we have performed additional experiments to understand the plasticity after stress with/without negative feedback that may be driving the changes in tonic activity. We replicated our protocol of using vehicle or metyrapone to manipulate endogenous negative feedback and prepared acute brain slices 1hr after white noise stress. Our new data (Figure 6) show that CORT feedback increases first spike latency in CRH neurons, likely mediated by a transient outward K⁺ current. We also show that while in vitro applications of CORT can reduce sEPSC frequency, this effect is not detected with our in vivo negative feedback model. We therefore believe that while CORT can suppress sEPSC frequency, the slow CORT induced suppression of tonic excitability in vivo is more likely to be mediated by a reduction in intrinsic excitability (also see response to comment 4 from reviewer 2).

Points for consideration:

1) In the first paragraph of the introduction, the authors state: “However, the plasticity mechanisms driving adaptations in these neural circuits following stress remain poorly studied”. A similar statement is made in line 44-45. I have two significant concerns here. First, to my mind this is not true, as the labs of Bains and Tasker, for example, have extensively studied plasticity in the PVN as it relates to stress and their work should be more explicitly mentioned in this context within the intro and discussion. Second, this statement seems to set up the notion that the authors will be shedding light on this issue in the current manuscript, yet I see no experiments that address plasticity per se in this system.

We would like to thank the reviewer for pointing this out. We have now made amendments to the introduction to correct any potentially misleading statements regarding plasticity.

2) The CORT resupply experiments are interesting, but discussion of the dosage chosen needs to be provided to give context on the physiologic versus supra-physiological relevance here.

We have cited previous work by the first author (using the same facility, the same ELISA and mouse strain) and others (using the same ELISA with mice) where an acute stress can raise blood CORT concentrations to ~300ng/ml. We have also added our own data following a 10 min restraint stress, which shows that CORT levels are elevated to 272 ng/mL. This has been added to the text (lines 483-487) and Supplementary Figure 3.

Our dose of 0.5mg/kg is 5 times lower than doses commonly used in other mouse studies (Kinlein et al, Psychoneuroendocrinology 2019; Liston and Gan, PNAS, 2011; Bahacek et al, Psychoneuroendocrinology 2015; Guarnieri et al, Biological Psychiatry 2012).

3) In Figure 3 the authors hypothesize a mechanism involving “presumably loss of

endogenous negative feedback” (line 151). This is the sort of statement that could be directly tested in a more mechanistically-oriented manuscript.

We have taken this comment on board in our revised manuscript and performed whole cell electrophysiology at the timepoint where we observe CORT effects on tonic CRH neuron activity in vivo. These experiments have revealed a lower level of intrinsic excitability and longer delay to first spike in animals where CORT negative feedback is intact. The difference in first spike latency between vehicle and metyrapone groups prompted us to also analyse the kinetics of spontaneous calcium transients measured in vivo (Figure 3I-L). We now report that the rising slope of calcium transients in metyrapone treated mice is significantly faster than vehicle controls. We speculate that this may be partly due to the shorter first spike latency seen in the metyrapone group.

We feel that these additions to our manuscript now provide more mechanistic insight.

4) Similarly lines 265-66 speculate on potential mechanisms that could be tested.

We have now addressed this particular statement by adding additional experiments.

5) In line 373, the authors propose CORT strongly inhibits the CRH stress response, as opposed to stating it desensitized or some other mechanism. Again, this could be directly tested.

We state that past experience of a familiar stress (but not CORT) strongly inhibits the CRH neuron response. We believe this is addressed in Figure 7 where two unfamiliar stressors elicit near-identical stress responses. Desensitization of neural excitability would suppress the second stressor (regardless of order), however, this was not observed.

6) In the description of the data in Figure 2H, there’s discussion of divergence of the curves at 3.5 minutes, but this is not indicated on the graph. Moreover, two bars with different asterisks are printed over the plots, but it’s not clear from the legend or results what they refer to. This presentation strategy is used throughout.

We apologise for this lack of clarity. We have added additional text to clarify this in the results (lines 136-139) and figure legends for all panels referring to this type of analysis.

7) The visual differences in the mean signals in Figure 3A seem heavily driven by just a few sweeps in the MET heatmap. The differences in these heatmaps seem somewhat subtle, and might be improved with a short timescale of presentation, since the analysis in 3C focuses on a short timescale.

We agree that the mean signals that show high elevations from baseline levels are driven by a small cohort of mice within the group. This effect is indeed subtle, however, we note that almost all animals exposed to negative feedback show tonic activity levels that is either the same or below baseline. Whereas animals without negative feedback show tonic activity that is either the same or above baseline. We have clarified this in the text (lines 150-155).

8) In Figure 3J, the far-right bar has two points that are very low relative to the others. Were the deposits and fiber placements verified in cases such as these?

We verified fibre placements in most of our mice but not all. However a more reliable method for verifying photometry recordings was to exclude any mice that showed stress induced peak signals that were below 25% $\Delta F/F$ from baseline. We did not include this criteria in our original manuscript and it has now been added (line 468). We also note that all mice served as their own internal controls to exclude variations in signal strengths as a confound between groups. This is described in lines 489 and 498.

In regard to the data/graph in Fig 3 J-L, the data has been normalised to peak WN values, which also removes signal strengths as a potential confound for such analysis. We have now clarified this in the figures and text.

9) The differences in Figure 4A through D are extremely subtle and one worries about

the influences of baseline drift, etc. Do the authors observe washout of this CORT effect?

We also had this same concern and this is why we took a long 40 min baseline period for the experiments in Figure 4. All linear regressions were based on the 405 slope which did not deviate from its original linear slope during the 40min baseline.

We did not observe a distinct washout of the CORT effect on tonic activity. This effect of CORT is theorised to be a genomic action and it is likely that such effects would be very long lasting. We note a recent publication (Beutler et al 2017 PMID: 29024666) where leptin has a similarly long lasting effect on POMC neuron activity that does not appear to wash out.

Minor issues:

1) Throughout, the authors simply refer to “CRH neurons”. This needs to be specified as hypothalamic CRH neurons given the heavy expression of this neuropeptide throughout extrahypothalamic cortical and subcortical structures responsive to stress.

We have now specified the term “hypothalamic CRH neurons” in the title. Due to the character limit of headings in the results section, we were unable to specify “Hypothalamic CRH neurons” here. In fact, most results headings in our revised manuscript have been markedly shortened to comply with the 60 character (including spaces) limit.

2) Line 108: “novel” would be more appropriate than “external” here.

This has now been changed.

3) Lines 216-17: Provide citations for these time windows.

We now provide a citation for this.

4) Line 261: sentence fragment on norepinephrine should be removed.

This has been corrected.

5) Line 374: “adaptive plasticity” is stated, but plasticity itself was not directly explored in this manuscript.

This has been amended.

Reviewer #4:

In this work, the authors use in vivo fiber photometry to record Ca²⁺ signals from GCaMP6s-expressing CRH neurons in the hypothalamus of animals exposed to a stressor (in this study very loud white noise or footshock). These Ca²⁺ signals can be used as a surrogate for neuronal action potential firing, and represent a mean population signal originating from all the somata, dendrites and axons located in the volume excited by the blue wavelength light. These experiments are very challenging, mainly because they were performed in freely-moving mice and because the hypothalamus is a very deep and hard-to-access brain structure. The in vivo fiber photometry experiments using two sinusoidally-modulated excitation wavelengths (405 nm being the wavelength where GCaMP fluorescence is Ca²⁺ insensitive and 465 nm the wavelength where GCaMP fluorescence is dependent on the cytosolic Ca⁺ concentration) are well designed and state-of-the-art. I have a few questions/comment that should be, in my opinion, addressed in a revised manuscript:

We thank for reviewer for their careful assessment of our manuscript. Responses to their specific comments are reported below.

1) The authors should clarify some basic properties of the fiber photometry system used here addressing the following questions: (1) What is the numerical aperture of the fiber used? (2) also, based on this feature, what is the (estimated) excitation volume of your system? (3) how many cells, including their axons/dendrites, are located in this volume?

The numerical aperture of the fiber is 0.48 and has been added to methods (lines 457-460). We estimate irradiance loss from 70 $\mu\text{W}/\text{mm}^2$ to 19 $\mu\text{W}/\text{mm}^2$ (61% power attenuation) within a distance of 0.2mm from the fibre tip. These values were calculated using resources from the Deisseroth lab website. The volume of tissue illuminated with light (up to 0.2mm from the fiber tip) is calculated to be 0.14263 mm^3 . This assumes a cone of light based on the NA of the fiber and the refractive index of brain tissue. However, there are several issues with this calculation including the fact that light emitted from a fiber optic in brain tissue is not a simple cone and secondly that the volume will vary greatly based on what threshold is set for the minimum level of illumination

(<https://www.biorxiv.org/content/biorxiv/early/2018/10/29/455766.full.pdf>)

We averaged total transfected neuron counts of 84.3 per 30 μm brain slice but did not quantify axon/dendrite volume. It is also important to note that the deepest transfected cells may not contribute much to the fiber photometry signal. For these reasons (and the ones noted above), we do not feel comfortable including statements in the manuscript in regard to the numbers of CRH cellular elements in the excitation volume.

2) In fig. S2, the authors show mean responses recorded from GCaMP6s- and GFP-expressing animals. The GFP expressing animals show almost no fluctuations. This result is evidence for the lack of any motion artifacts during the experiments, even when the animals are exposed to the stressor. However, these signals were collected from a different group of animals, and the number of data points is low (n=3). Instead, showing the 405 nm signal, the isosbestic wavelength of GCaMP6s, that was in any case recorded in the GCaMP6s-expressing animals would make the argument stronger. It would also be very informative since the 405 nm signal is used for the dF/F calculation. By the way, the authors mention the 405 nm reference channel in line 85 on page 5, but never show the actual data. Also, please clarify whether there was any data excluded due to motion artifacts.

We have added 405 nm signals to Supplemental Figure 2. We did not have to exclude any recordings due to movement artifacts. However, some mice were excluded prior to the start of experimentation if they showed photometry stress responses that were too small (peaks below 25% $\Delta\text{F}/\text{F}$ from baseline).

3) As shown, for example, in fig. 1j, the response to the white noise is biphasic and consists of an initial fast and higher-amplitude response and a sustained, but lower-amplitude response. Are these different “type” of responses caused by the same or different cell populations? Is there, for example, subsequent activation of new neurons by the initially responding neurons? It is interesting in this context that during repeated exposures over 4 days (figure 7) the peak response remains unchanged, whereas the sustained response is reduced.

This is an interesting point raised by the reviewer. We have not been able to identify particular subpopulations (fast vs slow or high vs low responders) in brain slices based on their anatomical positions within the PVN. A miniscope approach would address these concerns however we do not currently have this technique operational.

We theorize that the peak response is driven by neural inputs relaying startle responses. This is one possible reason why the initial peak response does not habituate like the sustained response.

4) Along the same lines: fig. 3i2 shows that after metyrapone application, there is an increase in the amplitude of the Ca²⁺ transients, but not in their frequency. It would be interesting to either experimentally address or at the very least discuss the possible mechanisms underlying this finding. Is there evidence for recruitment of neurons into the active neuron pool or do the same neurons “simply” fire more action potentials?

Our revised manuscript addresses this potential mechanism by conducting whole cell electrophysiology experiments at the timepoint at which these effects were observed. In brain slices from the metyrapone group, we find a higher level of intrinsic excitability and a shorter delay to first spike. The difference in first spike latency between vehicle and metyrapone groups prompted us to also analyse the kinetics of spontaneous calcium transients measured in vivo (Figure 3I-L). We now report that the rising slope of calcium transients in metyrapone treated mice is significantly faster than vehicle controls. We speculate that the higher level of intrinsic excitability will result in more spikes (hence larger event amplitude) and shorter delay to first spike will result in faster event rising slope.

Minor points:

1) Please clarify in the figure legends what animals are included in fig. 2g.

All animals shown in B are included in this trace. This has now been clarified.

2) It is not clear to me how the cumulative integrated dff, for example in fig. 2h, was computed. I would assume from the data that in the first 4 seconds after WN2 onset, there should be an increase in the cumulative integrated dff to 0.89.

The data for cumulative fluorescence is a sum of all values from individual animals, which is then group-averaged. Not the mean values from individual animals, which is then group-averaged.

3) page 8, line 180: I think this should be fig. 3H.

Thank you, this has been corrected.

Reviewers' comments:

Reviewer #1 (Remarks to the Author):

Authors addressed most of my comments, but there are a few issues that require further clarification as detailed below.

1) In authors' response #1, it was clarified that CORT administration prior to white noise (WN) exposure did not change CRH neurons' response during the 5-min exposure period. However, CORT decreased CRH neurons' activity during post-stress period (another 5 min). Accordingly, the overall (during stress + post-stress) cumulative CRH neurons' activity was decreased by 28%. This suppression is smaller than the magnitude of plasma ACTH suppression measured at the same time point (5 min post-stress, 63%). However, I am still not convinced that this direct comparison of the magnitudes of suppression supports the authors' conclusion that "CORT induced suppression of stress-evoked ACTH release is predominantly due to direct pituitary actions." (page 15, line 377-379). Also, in page 10, line 242-245, authors wrote "These results show that fast CORT negative feedback powerfully suppresses pituitary secretion while having a minor impact on the magnitude of CRH neuronal responses to threat.

I have two reasons why I am still not convinced. 1) The relationship between CRH neurons' activity and ACTH release is not linear considering the signaling cascades between the two (i.e. CRH release, metabotropic CRH receptor signaling, and ACTH release). Thus, the direct comparison of the magnitudes of suppression does not simply translate into the relative contribution of the CORT-induced suppression of CRH neuronal activity to the CORT-induced suppression of plasma ACTH elevation. 2) Considering that mice are pre-treated with metyrapone before CORT administration, Fig 5F ACTH measurement lacks an appropriate control (CORT + no WN). Metyrapone alone may increase the baseline ACTH levels, and CORT in the absence of WN may reduce the baseline ACTH levels. The lack of control makes it difficult to assess the effects of CORT on WN-induced ACTH release. For example, to calculate Δ ACTH (as done in the authors response #1) for CORT group, the baseline should be CORT + no WIN. This lack of important control has been raised in my original comment (Major comment #3) but not addressed in this revision. Terminal blood collection from CORT + no WIN mice is not a difficult experiment. Regarding the neural vs. pituitary mechanisms for fast CORT negative feedback, it is necessary to have more careful experimental designs and interpretations acknowledging the limitations.

2) More generally, my original point was asking to clarify the (or the lack of) relationship between distinct component of CRH neuronal activity and ACTH release. For example, does the slow actions of CORT that become evident > 30min post stress affect ACTH release? New data added for this revision (Fig. 6) shows that the slow actions of CORT decreases the intrinsic excitability of CRH neurons, and authors proposes that this mechanism contributes to the CORT-induced changes in the sustained CRH neurons activity. Considering this, this is an important point for this paper. One simple way to address this is to measure ACTH at 120 min after WN with or without metyrapone treatment (Fig. 3 experiments).

Minor comments:

1) Fig 2H (cumulative Δ F/F during and post-stress), and other similar graphs. It would be more informative to overlay gray shade (as in Fig. 2A) to indicate the period of WN (0-5min).

2) Fig. 6K, L. Two-way ANOVA should be used instead of one-way ANOVA.

Reviewer #2 (Remarks to the Author):

The authors have thoroughly and thoughtfully responded to my comments. I have no remaining concerns.

Reviewer #3 (Remarks to the Author):

The authors have done a great job addressing the issues I raised.

Reviewer #4 (Remarks to the Author):

All my comments have been addressed to my full satisfaction. I can now recommend this article for publications.

Response to reviewers.

We are pleased to see that we have addressed all of the concerns of Reviewers 2, 3 and 4. We are also glad to have addressed most of Reviewer 1's comments and hope we can address their remaining concerns below.

Reviewer #1:

Authors addressed most of my comments, but there are a few issues that require further clarification as detailed below.

1) In authors' response #1, it was clarified that CORT administration prior to white noise (WN) exposure did not change CRH neurons' response during the 5-min exposure period. However, CORT decreased CRH neurons' activity during post-stress period (another 5 min). Accordingly, the overall (during stress + post-stress) cumulative CRH neurons' activity was decreased by 28%. This suppression is smaller than the magnitude of plasma ACTH suppression measured at the same time point (5 min post-stress, 63%). However, I am still not convinced that this direct comparison of the magnitudes of suppression supports the authors' conclusion that "CORT induced suppression of stress-evoked ACTH release is predominantly due to direct pituitary actions." (page 15, line 377-379). Also, in page 10, line 242-245, authors wrote "These results show that fast CORT negative feedback powerfully suppresses pituitary secretion while having a minor impact on the magnitude of CRH neuronal responses to threat.

I have two reasons why I am still not convinced. 1) The relationship between CRH neurons' activity and ACTH release is not linear considering the signaling cascades between the two (i.e. CRH release, metabotropic CRH receptor signaling, and ACTH release). Thus, the direct comparison of the magnitudes of suppression does not simply translate into the relative contribution of the CORT-induced suppression of CRH neuronal activity to the CORT-induced suppression of plasma ACTH elevation. 2) Considering that mice are pre-treated with metyrapone before CORT administration, Fig 5F ACTH measurement lacks an appropriate control (CORT + no WN). Metyrapone alone may increase the baseline ACTH levels, and CORT in the absence of WN may reduce the baseline ACTH levels. The lack of control makes it difficult to assess the effects of CORT on WN-induced ACTH release. For example, to calculate Δ ACTH (as done in the authors response #1) for CORT group, the baseline should be CORT + no WIN. This lack of important control has been raised in my original comment (Major comment #3) but not addressed in this revision. Terminal blood collection from CORT + no WIN mice is not a difficult experiment. Regarding the neural vs. pituitary mechanisms for fast CORT negative feedback, it is necessary to have more careful experimental designs and interpretations acknowledging the limitations.

We agree with the reviewers first point that the relationship between CRH neuron activity and ACTH release may not be linear. Considering this and the fact that the overall focus of our manuscript is the effect of CORT on CRH neuron activity, we have modified the sentences in our manuscript to simply restate the data without drawing any neural vs pituitary mechanistic conclusions.

Line 377-379 has been replaced with: "Our current work shows that fast CORT feedback suppresses stress-evoked ACTH release 30 mins following injection. However, we observed only a subtle inhibition of stress-evoked CRH neuron activity during this period (Figure 5). While we did not observe any changes in basal CRH neuron activity in the first 30 min following CORT injection (Figure 4), we cannot rule out the possibility that CORT suppressed basal ACTH release during this period. Previous work has clearly shown that CORT negative feedback at the pituitary is important for the overall suppression of HPA axis output."

Line 242-245 has been replaced with: "These results show that fast CORT negative feedback suppresses ACTH secretion while having a minor impact on CRH neuron activity."

The reviewers second point centers around the idea that the most appropriate control group for our ACTH experiment would be "CORT + no WN". However, our ACTH experiment was designed to test how fast CORT negative feedback alters stress evoked ACTH release and not basal ACTH release. We agree with the reviewer that CORT may impact basal ACTH levels and have now raised this possibility in the discussion (see new manuscript text above).

2) More generally, my original point was asking to clarify the (or the lack of) relationship between distinct component of CRH neuronal activity and ACTH release. For example, does the slow actions of CORT that become evident > 30min post stress affect ACTH release? New data added for this revision (Fig. 6) shows that the slow actions of CORT decreases the intrinsic excitability of CRH neurons, and authors proposes that this mechanism contributes to the CORT-induced changes in the sustained CRH neurons activity. Considering this, this is an important point for this paper. One simple way to address this is to measure ACTH at 120 min after WN with or without metyrapone treatment (Fig. 3 experiments).

This is an interesting point. However, measuring ACTH at 120 min post stress will not discriminate between direct CORT effects on the pituitary versus effects on CRH neuron excitability (as noted by the reviewer in comment 1). As such, we do not believe that this experiment would provide additional insight into how CORT negative feedback regulates CRH neuron function.

Minor comments:

1) Fig 2H (cumulative $\Delta F/F$ during and post-stress), and other similar graphs. It would be more informative to overlay gray shade (as in Fig. 2A) to indicate the period of WN (0-5min).

We thank the reviewer for this helpful comment. This has now been added.

2) Fig. 6K, L. Two-way ANOVA should be used instead of one-way ANOVA.

We have re-run the analysis with Two-way ANOVA and the significant differences between the groups remain (albeit with slightly different p values).

REVIEWERS' COMMENTS:

Reviewer #1 (Remarks to the Author):

The authors have responded thoroughly to my comments. I now recommend this article for publication.